# Contextual Explanation Networks

## Abstract

We introduce *contextual explanation networks* (CENs)—a class of models that learn to predict by generating and leveraging intermediate explanations. CENs are deep networks that generate parameters for context-specific probabilistic graphical models which are further used for prediction and play the role of explanations. Contrary to the existing *post-hoc* model-explanation tools, CENs learn to predict and to explain jointly. Our approach offers two major advantages: (i) for each prediction, valid instance-specific explanations are generated with no computational overhead and (ii) prediction via explanation acts as a regularization and boosts performance in low-resource settings. We prove that local approximations to the decision boundary of our networks are consistent with the generated explanations. Our results on image and text classification and survival analysis tasks demonstrate that CENs are competitive with the state-of-the-art while offering additional insights behind each prediction, valuable for decision support.

## 1 Introduction

Model interpretability is a long-standing problem in machine learning that has become quite acute with the accelerating pace of widespread adoption of complex predictive algorithms. While high performance often supports our belief in predictive capabilities of a system, perturbation analysis reveals that black-box models can be easily broken in an unintuitive and unexpected manner (Szegedy et al., 2013; Nguyen et al., 2015). Therefore, for a machine learning system to be used in a social context (e.g., in healthcare) it is imperative to provide a sound reasoning for each decision.

Restricting the class of models to only *human-intelligible* (Caruana, 2015) is a potential remedy, but often is limiting in modern practical settings. Alternatively, we may fit a complex model and explain its predictions *post-hoc*, e.g., by searching for linear local approximations of the decision boundary (Ribeiro et al., 2016). While such approaches achieve their goal, the explanations are generated *a posteriori*, require additional computation per data instance, and most importantly are never the basis for the predictions made in the first place which may lead to erroneous interpretations.

Explanation is a fundamental part of the human learning and decision process (Lombrozo, 2006). Inspired by this fact, we introduce *contextual explanation networks* (CENs)—a class of deep neural networks that generate parameters for probabilistic graphical models. The generated models not only play the role of explanations but are used for prediction and can encode arbitrary prior knowledge. The data often consists of two representations: (1) low-level or unstructured features (e.g., text, image pixels, sensory inputs), and (2) high-level or human-interpretable features (e.g., categorical variables). To ensure interpretability, CENs use deep networks to process the low-level representation (called the *context*) and construct explanations as *context-specific probabilistic models* on the high-level features (*cf.* Koller & Friedman, 2009, Ch. 5.3). Importantly, the explanation mechanism is an integral part of CEN, and our models are trained to predict and to explain jointly.

**A motivating example.** Consider a CEN for diagnosing the risk of developing heart arrhythmia (Figure 1a). The causes of the condition are quite diverse, ranging from smoking and diabetes to an injury from previous heart attacks, and may carry different effects on the risk of arrhythmia in different contexts. Assume that the data for each patient consists of medical notes in the form of raw text (which is used as the *context*) and a number of specific attributes (such as high blood pressure, diabetes, smoking, etc.). Further, assume that we have access to a parametric class of expert-designed models that relate the attributes to the condition. The CEN maps the medical notes to the parameters of the model class to produce a *context-specific hypothesis*, which is further used to make a prediction.

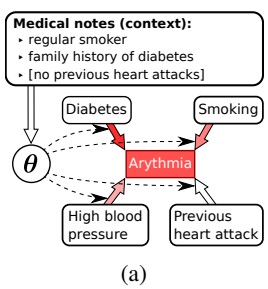 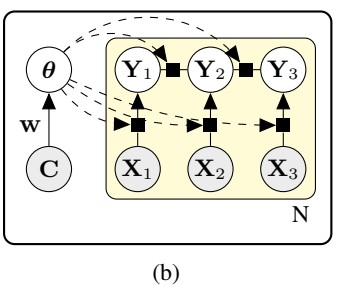 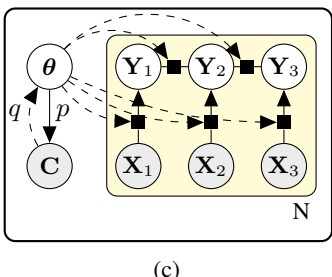

(a)               (b)               (c)

Figure 1: (a) An illustration of CEN for arrhythmia risk diagnosis. Shades of red denote the strength of association between the variables. (b) A graphical model for CEN with context encoder and CRF-based explanations. The model is parameterized by $\mathbf{w}$. (c) A graphical model for CEN with context autoencoding via the inference network, $q_{\mathbf{w}}(\boldsymbol{\theta} \mid \mathbf{C})$, generator network, $p_{\mathbf{u}}(\mathbf{C} \mid \boldsymbol{\theta})$, and CRF-based explanations.

In the sequel, we formalize these intuitions and refer to this toy example in our discussion to illustrate different aspects of the framework. The main contributions of the paper are as follows:

 (i) We formally define CENs as a class of probabilistic models, consider special cases (e.g., Jacobs et al., 1991), and derive learning and inference algorithms for simple and structured outputs.

 (ii) We prove that *post-hoc* approximations of CEN's decision boundary are consistent with the generated explanations and show that, in practice, while both methods tend to produce virtually identical explanations, CENs construct them orders of magnitude faster.

(iii) It turns out that noisy features can render *post-hoc* methods inconsistent and misleading, and we show how CENs can help to detect and avoid such situations.

(iv) We implement CENs by extending a number of established domain-specific deep architectures for image and text data and design new architectures for survival analysis. Experimentally, we demonstrate the value of learning with explanations for prediction and model diagnostics. Moreover, we find that explanations can act as a regularizer and improve sample efficiency.

## 2   RELATED WORK

**Deep graphical models.** The idea of combining deep networks with graphical models has been explored extensively. Notable threads of recent work include: replacing task-specific feature engineering with task-agnostic general representations (or embeddings) discovered by deep networks (Collobert et al., 2011; Rudolph et al., 2016; Rudolph et al., 2017), representing potential functions (Jaderberg et al., 2014) and energy functions (Belanger & McCallum, 2016) with neural networks, encoding learnable structure into Gaussian process kernels with deep and recurrent networks (Wilson et al., 2016; Al-Shedivat et al., 2017), or learning state-space models on top of nonlinear embeddings of observables (Gao et al., 2016; Johnson et al., 2016; Krishnan et al., 2017). The goal of this body of work is to design principled structured probabilistic models that enjoy the flexibility of deep learning. The key difference between CENs and previous art is that the latter directly integrate neural networks *into* graphical models as components (embeddings, potential functions, etc.). While flexible, the resulting *deep graphical models* could no longer be clearly interpreted in terms of crisp relationships between specific variables of interest[1]. CENs, on the other hand, preserve simplicity of the contextual models (explanations) and shift complexity into the process of conditioning on the context.

**Meta-learning.** The way CENs operate resembles the meta-learning setup. In meta-learning, the goal is to learn a meta-model which, given a task, can produce another model capable of solving the task (Thrun & Pratt, 1998). The representation of the task can be seen as the context while produced task-specific models are similar to CEN-generated explanations. Meta-training a deep network that generates parameters for another network has been successfully used for zero-shot (Lei Ba et al.,

---

[1]To see why this is the case, consider a simple graphical model given in Figure 1b that relates input variables, $\mathbf{X}$, to targets, $\mathbf{Y}$, using linear pairwise potential functions. Linearity allows us to directly interpret parameters of the model as associations between pairs of variables. Substituting inputs, $\mathbf{X}$, with deep latent representations, $\mathbf{H}$, or representing pairwise potentials with neural networks would result in a more powerful model. However, precise relationships between $\mathbf{X}$ and $\mathbf{Y}$ variables will be no longer directly readable from the model parameters.

2015; Changpinyo et al., 2016) and few-shot (Edwards & Storkey, 2016; Vinyals et al., 2016) learning, cold-start recommendations (Vartak et al., 2017), and a few other scenarios (Bertinetto et al., 2016; De Brabandere et al., 2016; Ha et al., 2016), but is not suitable for interpretability purposes. In contrast, CENs generate parameters for models from a restricted class (potentially, based on domain knowledge) and use the attention mechanism (Xu et al., 2015) to further improve interpretability. Using explanations based on domain knowledge is known to improve generalization (Mitchell et al., 1986) and could be used as a powerful mechanism for solving complex downstream tasks such as program induction for solving algebraic word problems (Ling et al., 2017).

**Context representation.** Generating a probabilistic model by conditioning on a context is the key aspect of our approach. Previous work on context-specific graphical models represented contexts with a discrete variable that enumerated a finite number of possible contexts (*cf.* Koller & Friedman, 2009, Ch. 5.3). CENs, on the other hand, are designed to handle arbitrary complex context representations. We also note that context-specific approaches are widely used in language modeling where the context is typically represented with trainable embeddings (Rudolph et al., 2016; Liu et al., 2017).

**Interpretability.** While there are many ways to define interpretability (Lipton, 2016; Doshi-Velez & Kim, 2017), our discussion focuses on explanations defined as simple models that locally approximate behavior of a complex model. A few methods that allow to construct such explanations in a *post-hoc* manner have been proposed recently (Ribeiro et al., 2016; Shrikumar et al., 2017; Lundberg & Lee, 2017). In contrast, CENs learn to generate such explanations along with predictions. There are multiple other complementary approaches to interpretability ranging from a variety of visualization techniques (Simonyan & Zisserman, 2014; Yosinski et al., 2015; Mahendran & Vedaldi, 2015; Karpathy et al., 2015), to explanations by example (Caruana et al., 1999; Kim et al., 2014; Kim et al., 2016; Koh & Liang, 2017), to natural language rationales (Lei et al., 2016). Finally, our framework encompasses the class of so-called *personalized* or *instance-specific* models that learn to partition the space of inputs and fit local sub-models (Wang & Saligrama, 2012).

## 3 METHODS

We consider the problem of learning from a collection of data where each instance is represented by three random variables: the *context*, $\mathbf{C} \in \mathcal{C}$, the *attributes*, $\mathbf{X} \in \mathcal{X}$, and the *targets*, $\mathbf{Y} \in \mathcal{Y}$. Our goal is to learn a model, $p_{\mathbf{w}}(\mathbf{Y} \mid \mathbf{X}, \mathbf{C})$, parametrized by $\mathbf{w}$ that can predict $\mathbf{Y}$ from $\mathbf{X}$ and $\mathbf{C}$. We define contextual explanation networks as models that assume the following form (Figure 1b):

$$\mathbf{Y} \sim p(\mathbf{Y} \mid \mathbf{X}, \boldsymbol{\theta}), \quad \boldsymbol{\theta} \sim p_{\mathbf{w}}(\boldsymbol{\theta} \mid \mathbf{C}), \quad p_{\mathbf{w}}(\mathbf{Y} \mid \mathbf{X}, \mathbf{C}) = \int p(\mathbf{Y} \mid \mathbf{X}, \boldsymbol{\theta}) p_{\mathbf{w}}(\boldsymbol{\theta} \mid \mathbf{C}) d\boldsymbol{\theta} \quad (1)$$

where $p(\mathbf{Y} \mid \mathbf{X}, \boldsymbol{\theta})$ is a predictor parametrized by $\boldsymbol{\theta}$. We call such predictors *explanations*, since they explicitly relate interpretable variables, $\mathbf{X}$, to the targets, $\mathbf{Y}$. For example, when the targets are scalar and binary, explanations may take the form of linear logistic models; when the targets are more complex, dependencies between the components of $\mathbf{Y}$ can be represented by a graphical model, e.g., a conditional random field (Lafferty et al., 2001).

CENs assume that each explanation is context-specific: $p_{\mathbf{w}}(\boldsymbol{\theta} \mid \mathbf{C})$ defines a conditional probability of an explanation $\boldsymbol{\theta}$ being valid in the context $\mathbf{C}$. To make a prediction, we marginalize out $\boldsymbol{\theta}$'s; to interpret a prediction, $\mathbf{Y} = \mathbf{y}$, for a given data instance, $(\mathbf{x}, \mathbf{c})$, we infer the posterior, $p_{\mathbf{w}}(\boldsymbol{\theta} \mid \mathbf{Y} = \mathbf{y}, \mathbf{x}, \mathbf{c})$. The main advantage of this approach is to allow modeling conditional probabilities, $p_{\mathbf{w}}(\boldsymbol{\theta} \mid \mathbf{C})$, in a black-box fashion while keeping the class of explanations, $p(\mathbf{Y} \mid \mathbf{X}, \boldsymbol{\theta})$, simple and interpretable. For instance, when the context is given as raw text, we may choose $p_{\mathbf{w}}(\boldsymbol{\theta} \mid \mathbf{C})$ to be represented with a recurrent neural network, while $p(\mathbf{Y} \mid \mathbf{X}, \boldsymbol{\theta})$ be in the class of linear models.

Implications of the assumptions made by (1) are discussed in Appendix A. Here, we move on to describing a number of practical choices for $p_{\mathbf{w}}(\boldsymbol{\theta} \mid \mathbf{C})$ and learning and inference for those.

### 3.1 CONTEXTUAL EXPLANATION NETWORKS

In practice, we represent $p_{\mathbf{w}}(\boldsymbol{\theta} \mid \mathbf{C})$ with a neural network that encodes the context into the parameter space of explanations. There are multiple ways to construct an encoder, which we consider below.

**Deterministic Encoding.** Consider $p_{\mathbf{w}}(\boldsymbol{\theta} \mid \mathbf{C}) := \delta\left(\phi_{\mathbf{w}}(\mathbf{C}), \boldsymbol{\theta}\right)$, where $\delta(\cdot, \cdot)$ is a delta-function and $\phi_{\mathbf{w}}$ is the network that maps $\mathbf{C}$ to $\boldsymbol{\theta}$. Collapsing the conditional distribution to a delta-function makes

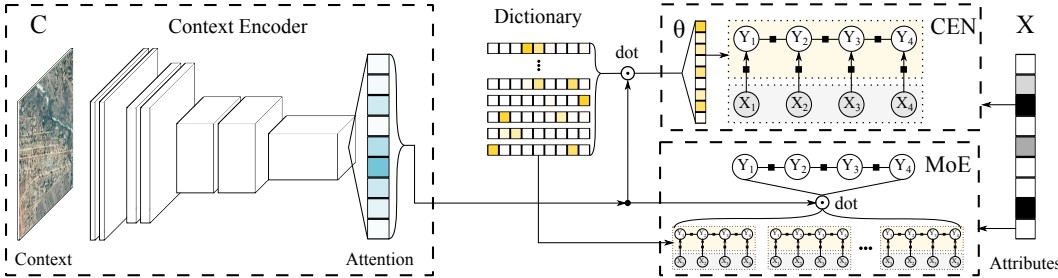

Figure 2: An example of CEN architecture. The context is represented by an image and transformed by a convnet encoder into an attention vector, which is used to construct a contextual hypothesis from a dictionary of sparse atoms. MoE uses a similar attention mechanism but for combining predictions of each model in the dictionary.

$\boldsymbol{\theta}$ depend deterministically on $\mathbf{C}$ and results into the following tractable conditional log-likelihood:

$$\log p(\mathbf{y}_i \mid \mathbf{x}_i, \mathbf{c}_i; \mathbf{w}) = \log \int p(\mathbf{y}_i \mid \mathbf{x}_i, \boldsymbol{\theta})\delta\left(\phi_{\mathbf{w}}(\mathbf{c}_i), \boldsymbol{\theta}\right) d\boldsymbol{\theta} = \log p(\mathbf{y}_i \mid \mathbf{x}_i, \boldsymbol{\theta}_i = \phi_{\mathbf{w}}(\mathbf{c}_i)) \quad (2)$$

Since $p_{\mathbf{w}}(\boldsymbol{\theta}_i \mid \mathbf{y}_i, \mathbf{x}_i, \mathbf{c}_i) \propto p(\mathbf{y}_i \mid \mathbf{x}_i, \boldsymbol{\theta}_i)\delta\left(\phi_{\mathbf{w}}(\mathbf{c}_i), \boldsymbol{\theta}_i\right)$, the posterior also collapses to $\boldsymbol{\theta}_i^{\star} = \phi_{\mathbf{w}}(\mathbf{c}_i)$, and hence the inference is done via a single forward pass.

**Constrained Deterministic Encoding.** The downside of deterministic encoding is the lack of constraints on the generated explanations. There are multiple reasons why this might be an issue: (i) when the context encoder is unrestricted, it might generate unstable, overfitted local models, (ii) explanations are not guaranteed to be human-interpretable *per se*, and often require imposing additional constraints, such as sparsity, and (iii) when we want to reason about the patterns in the data as a whole, local explanations are not enough. To address these issues, we constrain the space of explanations by introducing a *global dictionary*, $\mathbf{D} := \{\boldsymbol{\theta}_k\}_{k=1}^{K}$, where each atom of the dictionary, $\boldsymbol{\theta}_k$, is sparse. The encoder generates context-specific explanations using *soft attention* over the dictionary, i.e., each explanation becomes a convex combination of the sparse atoms (Figure 2):

$$\phi_{\mathbf{w},\mathbf{D}}(\mathbf{c}) = \sum_{k=1}^{K} p_{\mathbf{w}}(k \mid \mathbf{c})\boldsymbol{\theta}_k = \boldsymbol{\alpha}_{\mathbf{w}}(\mathbf{c})^{\top}\mathbf{D}, \quad \sum_{k=1}^{K} \boldsymbol{\alpha}_{\mathbf{w}}^{(k)}(\mathbf{c}) = 1, \quad \forall k : \boldsymbol{\alpha}_{\mathbf{w}}^{(k)}(\mathbf{c}) \geq 0 \quad (3)$$

where $\boldsymbol{\alpha}_{\mathbf{w}}(\mathbf{c})$ is the attention over the dictionary. As previously, the encoder is a delta-distribution, $p_{\mathbf{w},\mathbf{D}}(\boldsymbol{\theta} \mid \mathbf{C}) := \delta\left(\phi_{\mathbf{w},\mathbf{D}}(\mathbf{C}), \boldsymbol{\theta}\right)$. The model is trained by learning the weights, $\mathbf{w}$ and the dictionary, $\mathbf{D}$. The log-likelihood is as given in (2), and learning and inference are done via a forward pass[2].

**Mixtures of Experts.** So far, we represented $p_{\mathbf{w}}(\boldsymbol{\theta} \mid \mathbf{C})$ by a delta-function centered around the output of the encoder. It is natural to extend $p_{\mathbf{w}}(\boldsymbol{\theta} \mid \mathbf{C})$ to a mixture of delta-distributions, in which case CENs recover the *mixtures of experts* (MoE, Jacobs et al., 1991). In particular, let $\mathbf{D} := \{\boldsymbol{\theta}_k\}_{k=1}^{K}$ be now a dictionary of experts, and define the encoder as $p_{\mathbf{w},\mathbf{D}}(\boldsymbol{\theta} \mid \mathbf{C}) = \sum_{k=1}^{K} p_{\mathbf{w}}(k \mid \mathbf{C})\delta(\boldsymbol{\theta}, \boldsymbol{\theta}_k)$. The log-likelihood in such case is the same as for MoE:

$$\log p_{\mathbf{w},\mathbf{D}}(\mathbf{y}_i \mid \mathbf{x}_i, \mathbf{c}_i) = \log \int p(\mathbf{y}_i|\mathbf{x}_i, \boldsymbol{\theta})p_{\mathbf{w},\mathbf{D}}(\boldsymbol{\theta}|\mathbf{c}_i)d\boldsymbol{\theta} = \log \sum_{k=1}^{K} p_{\mathbf{w}}(k|\mathbf{c}_i)p(\mathbf{y}_i|\mathbf{x}_i, \boldsymbol{\theta}_k) \quad (4)$$

Note that $p_{\mathbf{w}}(k \mid \mathbf{C})$ is also represented as soft attention over the dictionary, $\mathbf{D}$, which is now used for combining predictions of each expert, $\boldsymbol{\theta}_k$, for a given context, $\mathbf{C}$, instead of constructing a single context-specific explanation. Learning is done by either directly optimizing the log-likelihood (4) or via EM. To infer an explanation for a given context, we compute the posterior (see Appendix C).

**Contextual Variational Autoencoders.** Modeling $p(\mathbf{Y} \mid \mathbf{X}, \mathbf{C})$ in the form of (1) avoids representing the joint distribution, $p(\boldsymbol{\theta}, \mathbf{C})$, which is a good decision when the data is abundant. However, incorporating a generative model of the context provides a few benefits: (i) a better regularization in low-resource settings, and (ii) a coherent Bayesian framework that allows imposing additional priors

---

[2]Note that deterministic encoding and the dictionary constraint assume that all explanations have the same graphical structure and parameterization. Having a more hierarchical or structured space of explanations should be possible using ideas from amortized inference (Rudolph et al., 2017). We leave this direction to future work.

on the parameters of explanations, $\boldsymbol{\theta}$. We accomplish this by representing $p(\boldsymbol{\theta}, \mathbf{C})$ with a variational autoencoder (VAE) (Kingma & Welling, 2013; Rezende et al., 2014) whose latent variables are explanation parameters (Figure 1c). The generative process and the evidence lower bound (ELBO) are as follows:

$$
\begin{aligned}
&\boldsymbol{\theta} \sim p_{\boldsymbol{\gamma}}(\boldsymbol{\theta}), \ \mathbf{C} \sim p_{\mathbf{u}}(\mathbf{C} \mid \boldsymbol{\theta}), \ \mathbf{Y} \sim p(\mathbf{Y} \mid \mathbf{X}, \boldsymbol{\theta}), \\
&\log p(\mathbf{Y}, \mathbf{C} \mid \mathbf{X}) \geq \mathbb{E}_{q_{\mathbf{w}}(\boldsymbol{\theta} \mid \mathbf{C})} \left[ \log p(\mathbf{Y}, \mathbf{C} \mid \mathbf{X}, \boldsymbol{\theta}) \right] - \mathrm{KL} \left( q_{\mathbf{w}}(\boldsymbol{\theta} \mid \mathbf{C}) \| \, p(\boldsymbol{\theta}) \right)
\end{aligned}
\tag{5}
$$

where $p(\mathbf{Y}, \mathbf{C} \mid \mathbf{X}, \boldsymbol{\theta}) := p(\mathbf{Y} \mid \mathbf{X}, \boldsymbol{\theta}) p_{\mathbf{u}}(\mathbf{C} \mid \boldsymbol{\theta})$, and $q_{\mathbf{w}}(\boldsymbol{\theta} \mid \mathbf{C})$ and $p_{\mathbf{u}}(\mathbf{C} \mid \boldsymbol{\theta})$ the encoder and decoder, respectively. We consider encoders that also make use of the global learnable dictionary, $\mathbf{D}$, and represent $q_{\mathbf{w}}(\boldsymbol{\theta} \mid \mathbf{C})$ in the form of logistic normal distribution over the simplex spanned by the atoms of $\mathbf{D}$. For the prior, $p_{\boldsymbol{\gamma}}(\boldsymbol{\theta})$, we use a Dirichlet distribution with parameters $\alpha_k < 1$ to induce sharp attention. Derivations are deferred to Appendix D.

## 3.2 CEN-GENERATED VS. POST-HOC EXPLANATIONS

In this section, we analyze the relationship between CEN-generated and LIME-generated *post-hoc* explanations. LIME (Ribeiro et al., 2016) constructs explanations as local linear approximations of the decision boundary of a model $f$ in the neighborhood of a given point $(\mathbf{x}, \mathbf{c})$ via optimization:

$$
\hat{\boldsymbol{\theta}} = \underset{\boldsymbol{\theta}}{\mathrm{argmin}} \ \mathcal{L}(f, \boldsymbol{\theta}, \pi_{\mathbf{x}, \mathbf{c}}) + \Omega(\boldsymbol{\theta}),
\tag{6}
$$

where $\mathcal{L}(f, \boldsymbol{\theta}, \pi_{\mathbf{x}, \mathbf{c}})$ measures the quality of the linear model $g_{\boldsymbol{\theta}} : \mathbf{X} \mapsto \mathbf{Y}$ as an approximation to $f$ in the neighborhood of $(\mathbf{x}, \mathbf{c})$, and $\Omega(\boldsymbol{\theta})$ is a regularizer. The typical choice for $\mathcal{L}$ and $\Omega$ is $L_2$ and $L_1$ losses, respectively. The neighborhood of $(\mathbf{x}, \mathbf{c})$ is defined by a distribution $\pi_{\mathbf{x}, \mathbf{c}}$ concentrated around the point of interest. Given a trained CEN, we can use LIME to approximate its decision boundary and compare the explanations produced by both methods. The question we ask:

> *How does the local approximation, $\hat{\boldsymbol{\theta}}$, relate to the actual explanation, $\boldsymbol{\theta}^\star$, generated and used by CEN to make a prediction in the first place?*

For the case of binary[3] classification, it turns out that when the context encoder is deterministic and the space of explanations is *linear*, local approximations, $\hat{\boldsymbol{\theta}}$, obtained by solving (6) recover the original CEN-generated explanations, $\boldsymbol{\theta}^\star$. Formally, our result is stated in the following theorem.

**Theorem 1.** *Let the explanations and the local approximations be in the class of linear models, $p(Y = 1 \mid \mathbf{x}, \boldsymbol{\theta}) \propto \exp \left\{ \mathbf{x}^\top \boldsymbol{\theta} \right\}$. Further, let the encoder be L-Lipschitz and pick a sampling distribution, $\pi_{\mathbf{x}, \mathbf{c}}$, that concentrates around the point $(\mathbf{x}, \mathbf{c})$, such that $p_{\pi_{\mathbf{x}, \mathbf{c}}}(\|\mathbf{z}' - \mathbf{z}\| > t) < \varepsilon(t)$, where $\mathbf{z} := (\mathbf{x}, \mathbf{c})$ and $\varepsilon(t) \to 0$ as $t \to \infty$. Then, if the loss function is defined as*

$$
\mathcal{L} = \frac{1}{K} \sum_{k=1}^{K} \left( \mathrm{logit} \, p(Y = 1 \mid \mathbf{x}_k, \mathbf{c}_k) - \mathrm{logit} \, p(Y = 1 \mid \mathbf{x}_k, \boldsymbol{\theta}) \right)^2, \ (\mathbf{x}_k, \mathbf{c}_k) \sim \pi_{\mathbf{x}, \mathbf{c}},
\tag{7}
$$

*the solution of (6) concentrates around $\boldsymbol{\theta}^\star$ as $\mathbb{P}_{\pi_{\mathbf{x}, \mathbf{c}}} \left( \|\hat{\boldsymbol{\theta}} - \boldsymbol{\theta}^\star\| > t \right) \leq \delta_{K, L}(t), \text{ for } \delta_{K, L} \xrightarrow[t \to \infty]{} 0.$*

Intuitively, by sampling from a distribution sharply concentrated around $(\mathbf{x}, \mathbf{c})$, we ensure that $\hat{\boldsymbol{\theta}}$ will recover $\boldsymbol{\theta}^\star$ with high probability. The proof is given in Appendix B.

This result establishes an equivalence between the explanations generated by CEN and those produced by LIME *post-hoc* when approximating CEN. Note that when LIME is applied to a model other than CEN, equivalence between explanations is not guaranteed. Moreover, as we further show experimentally, certain conditions such as incomplete or noisy interpretable features may lead to LIME producing inconsistent and erroneous explanations.

## 3.3 STRUCTURED EXPLANATIONS FOR SURVIVAL TIME PREDICTION

While CEN and LIME generate similar explanations in the case of simple classification (i.e., when $\mathbf{Y}$ is a scalar), when $\mathbf{Y}$ is structured (e.g., as a sequence), constructing coherent local approximations

---

[3]Analysis of the multi-class case can be reduced to the binary in the one-vs-all fashion.

in a *post-hoc* manner is non-trivial. At the same time, CENs naturally let us represent $p(\mathbf{Y} \mid \mathbf{X}, \boldsymbol{\theta})$ using arbitrary graphical models. To demonstrate our approach, we consider survival time prediction task where interpretability can be uniquely valuable (e.g., in a medical setting). Survival analysis can be re-formulated as a sequential prediction problem (Yu et al., 2011). To this end, we design CENs with CRF-based explanations suitable for sequentially structured outputs.

Our setup is as follows. Again, the data instances are represented by contexts, $\mathbf{C}$, attributes, $\mathbf{X}$, and targets, $\mathbf{Y}$. The difference is that now targets are sequences of $m$ binary variables, $\mathbf{Y} := (y^1, \ldots, y^m)$, that indicate occurrence of an event. If the event occurred at time $t \in [t_i, t_{i+1})$, then $y^j = 0$, $\forall j \leq i$ and $y^k = 1$, $\forall k > i$. If the event was *censored* (i.e., we lack information for times after $t \in [t_i, t_{i+1})$), we represent targets $(y^{i+1}, \ldots, y^m)$ with latent variables. Note that only $m + 1$ sequences are valid, i.e., assigned non-zero probability by the model. We define CRF-based CEN as:

$$\boldsymbol{\theta}^t \sim p_{\mathbf{w}}(\boldsymbol{\theta}^t \mid \mathbf{C}), \ t \in \{1, \ldots, m\}, \quad \mathbf{Y} \sim p(\mathbf{Y} \mid \mathbf{X}, \boldsymbol{\theta}^{1:m}),$$

$$p(\mathbf{Y} = (y^1, y^2, \ldots, y^m) \mid \mathbf{x}, \boldsymbol{\theta}^{1:m}) \propto \exp\left\{\sum_{t=1}^{m} y^i(\mathbf{x}^\top \boldsymbol{\theta}^t) + \omega(y^t, y^{t+1})\right\} \quad (8)$$

$$p_{\mathbf{w}}(\boldsymbol{\theta}^t \mid \mathbf{C}) := \delta(\boldsymbol{\theta}^t, \phi_{\mathbf{w},\mathbf{D}}^t(\mathbf{c})), \quad \phi_{\mathbf{w},\mathbf{D}}^t(\mathbf{c}) := \boldsymbol{\alpha}(\mathbf{h}^t)^\top \mathbf{D}, \quad \mathbf{h}^t := \mathrm{RNN}(\mathbf{h}^{t-1}, \mathbf{c})$$

Note that here we have explanations for each time point, $\boldsymbol{\theta}^{1:m}$, and use an RNN-based encoder $\phi^t$. The potentials between attributes, $\mathbf{x}$, and targets, $y^{1:m}$, are linear functions parameterized by $\boldsymbol{\theta}^{1:m}$; the pairwise potentials between targets, $\omega(y_i, y_{i+1})$, ensure that configurations $(y_i = 1, y_{i+1} = 0)$ are improbable (i.e., $\omega(1, 0) = -\infty$ and $\omega(0, 0) = \omega_{00}$, $\omega(0, 1) = \omega_{01}$, $\omega(1, 1) = \omega_{10}$ are learnable parameters). Given these constraints, the likelihood of an uncensored event at time $t \in [t_j, t_{j+1})$ is

$$p(T = t \mid \mathbf{x}, \boldsymbol{\Theta}) = \exp\left\{\sum_{i=j}^{m} \mathbf{x}^\top \boldsymbol{\theta}^i\right\} \Bigg/ \sum_{k=0}^{m} \exp\left\{\sum_{i=k+1}^{m} \mathbf{x}^\top \boldsymbol{\theta}^i\right\} \quad (9)$$

and the likelihood of an event censored at time $t \in [t_j, t_{j+1})$ is

$$p(T \geq t \mid \mathbf{x}, \boldsymbol{\Theta}) = \sum_{k=j+1}^{m} \exp\left\{\sum_{i=k+1}^{m} \mathbf{x}^\top \boldsymbol{\theta}^i\right\} \Bigg/ \sum_{k=0}^{m} \exp\left\{\sum_{i=k+1}^{m} \mathbf{x}^\top \boldsymbol{\theta}^i\right\} \quad (10)$$

The joint log-likelihood of the data consists of two parts: (a) the sum over the non-censored instances, for which we compute $\log p(T = t \mid \mathbf{x}, \boldsymbol{\Theta})$, and (b) sum over the censored instances, for which we use $\log p(T \geq t \mid \mathbf{x}, \boldsymbol{\Theta})$. We provide a much more elaborate discussion of the survival time prediction setup and our architectures in Appendix E.

## 4 EXPERIMENTS

For empirical evaluation, we consider applications that involve different data modalities of the context: image, text, and time-series. In each case, CENs are based on deep architectures designed for learning from the given type of context. In the first part, we focus on classification tasks and use linear logistic models as explanations. In the second part, we apply CENs to survival analysis and use structured explanations in the form of conditional random fields (CRFs).

We design our experiments around the following questions:

  (i) When explanation is a part of the learning and prediction process, how does that affect performance of the predictive model? Does the learning become more or less efficient both in terms of convergence and sample complexity? How do CENs stand against vanilla deep nets?

 (ii) Explanations are as good as the features they use to explain predictions. We ask how noisy interpretable features affect explanations generated *post-hoc* by LIME and whether CEN can help to detect and avoid such situations.

(iii) Finally, we ask what kind of insight we can gain by visualizing and inspecting explanations?

Details on the setup, all hyperparameters, and training procedures are given in Appendices F.1 and F.3.

Table 1: Performance of the models on classification tasks (averaged over 5 runs; the std. are on the order of the least significant digit). The subscripts denote the features on which the linear models are built: pixels (pxl), HOG (hog), bag-or-words (bow), topics (tpc), embeddings (emb), discrete attributes (att).

| MNIST | | CIFAR10 | | IMDB | | Satellite | | |
|---|---|---|---|---|---|---|---|---|
| Model | Err (%) | Model | Err (%) | Model | Err (%) | Model | Acc (%) | AUC (%) |
| $LR_{pxl}$ | 8.00 | $LR_{pxl}$ | 60.1 | $LR_{bow}$ | 13.3 | $LR_{emb}$ | 62.5 | 68.1 |
| $LR_{hog}$ | 2.98 | $LR_{hog}$ | 48.6 | $LR_{tpc}$ | 17.1 | $LR_{att}$ | 75.7 | 82.2 |
| CNN | **0.75** | VGG | 9.4 | LSTM | 13.2 | MLP | 77.4 | 78.7 |
| $MoE_{pxl}$ | 1.23 | $MoE_{pxl}$ | 13.0 | $MoE_{bow}$ | 13.9 | $MoE_{att}$ | 77.9 | **85.4** |
| $MoE_{hog}$ | 1.10 | $MoE_{hog}$ | 11.7 | $MoE_{tpc}$ | 12.2 | $CEN_{att}$ | 81.5 | 84.2 |
| $CEN_{pxl}$ | 0.76 | $CEN_{pxl}$ | 9.6 | $CEN_{bow}$ | $^\star$**6.9** | $VCEN_{att}$ | **83.4** | 84.6 |
| $CEN_{hog}$ | **0.73** | $CEN_{hog}$ | **9.2** | $CEN_{tpc}$ | $^\star$7.8 | | | |

$^\star$Best previous results for supervised learning and similar LSTM architectures: 8.1% (Johnson & Zhang, 2016).

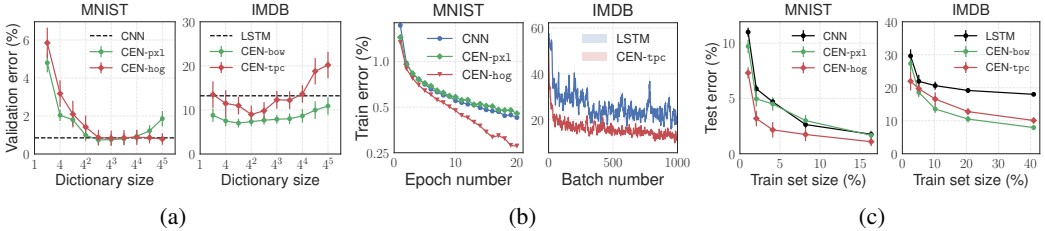

Figure 3: (a) Validation error vs. the size of the dictionary. (b) Training error vs. iteration (epoch or batch) for baselines and CENs. (c) Test error for models trained on random subsets of data of different sizes.

## 4.1 CLASSIFICATION TASKS AND LINEAR EXPLANATIONS

**Classical datasets.** We consider two classical image datasets, MNIST[4] and CIFAR10[5], and a text dataset for sentiment classification of IMDB reviews (Maas et al., 2011). For MNIST and CIFAR10: full images are used as the context; to imitate high-level features, we use (a) the original images cubically downscaled to $20 \times 20$ pixels, gray-scaled and normalized, and (b) HOG descriptors computed using $3 \times 3$ blocks (Dalal & Triggs, 2005). For IMDB: the context is represented by sequences of words; for high-level features we use (a) the bag-of-words (BoW) representation and (b) the 50-dimensional topic representation produced by a separately trained off-the-shelf topic model. Neither data augmentation, nor pre-training or other unsupervised techniques were used.

**Remote sensing.** We also consider the problem of poverty prediction for household clusters in Uganda from satellite imagery and survey data (the dataset is referred to as *Satellite*). Each household cluster is represented by a collection of $400 \times 400$ satellite images (used as the context) and 65 categorical variables from living standards measurement survey (used as the interpretable attributes). The task is binary classification of the households into poor and not poor. We follow the original study of Jean et al. (2016) and use a pre-trained VGG-F network to compute 4096-dimensional embeddings of the satellite images on top of which we build contextual models. Note that this datasets is fairly small (642 points), and hence we keep the VGG-F part of the model frozen to avoid overfitting.

**Models.** For each task, we use linear regression and vanilla deep nets as baselines. For MNIST and CIFAR10, the networks are a simple convnet (2 convolutions followed by max pooling) and the VGG-16 architecture (Simonyan & Zisserman, 2014), respectively. For IMDB, following Johnson & Zhang (2016) and we use a bi-directional LSTM with max pooling. For Satellite, we use a fixed VGG-F followed by a multi-layer perceptron (MLP) with 1 hidden layer. Our models used the baseline deep architectures as their context encoders and were of three types: (a) CENs with constrained deterministic encoding (b) mixture of experts (MoE), (c) CENs with variational context autoencoding (VCEN). All our models use the dictionary constraint and sparsity regularization.

---

[4] http://yann.lecun.com/exdb/mnist/
[5] http://www.cs.toronto.edu/~kriz/cifar.html

### 4.1.1 EXPLANATIONS AS A REGULARIZER

In this part, we compare CENs with the baselines in terms of performance. In each task, CENs are trained to simultaneously generate predictions and construct explanations using a global dictionary. When the dictionary size is 1, they become equivalent to linear models. For larger dictionaries, CENs become as flexible as deep nets (Figure 3a). Adding a small sparsity penalty on the dictionary (between $10^{-6}$ and $10^{-3}$, see Tables 3, 4, 5) helps to avoid overfitting for very large dictionary sizes, so that the model learns to use only a few dictionary atoms for prediction while shrinking the rest to zeros. Overall, CENs show very competitive performance and are able to approach or surpass baselines in a number of cases, especially on the IMDB data (Table 1). Thus, forcing the model to produce explanations along with predictions does not limit its capacity.

Additionally, the "explanation layer" in CENs somehow affects the geometry of the optimization problem, and we notice that it often causes faster convergence (Figure 3b). When the models are trained on a subset of data (size varied between 1% and 20% for MNIST and 2% and 40% for IMDB), explanations play the role of a regularizer which strongly improves the sample efficiency of our models (Figure 3c). This becomes even more evident from the results on the Satellite dataset that had only 500 training points: contextual explanation networks significantly improved upon the sparse linear models on the survey features (known as the gold standard in remote sensing). Note that training an MLP on both the satellite image features and survey variables, while beneficial, does not come close to the result achieved by contextual explanation networks (Table 1).

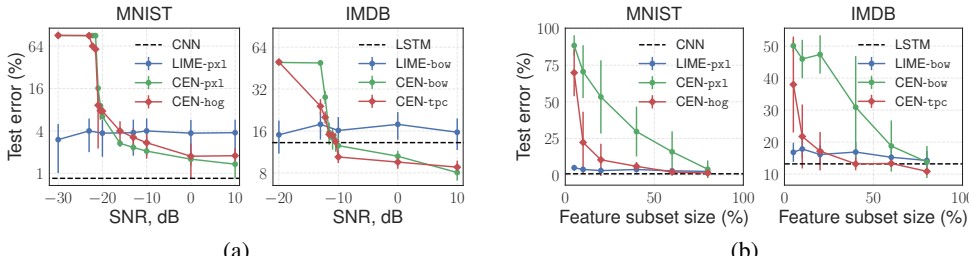

Figure 4: The effect of feature quality on explanations. (a) Explanation test error vs. the level of the noise added to the interpretable features. (b) Explanation test error vs. the total number of interpretable features.

### 4.1.2 CONSISTENCY OF EXPLANATIONS

While regularization is a useful aspect, the main use case for explanations is model diagnostics. Linear explanation assign weights to the interpretable features, $\mathbf{X}$, and hence their quality depends on the way we select these features. We consider two cases where (a) the features are corrupted with additive noise, and (b) the selected features are incomplete. For analysis, we use MNIST and IMDB datasets. Our question is, *Can we trust the explanations on noisy or incomplete features?*

**The effect of noisy features.** In this experiment, we inject noise[6] into the features $\mathbf{X}$ and ask LIME and CEN to fit explanations to the corrupted features. Note that after injecting noise, each data point has a noiseless representation $\mathbf{C}$ and noisy $\mathbf{X}$. LIME constructs explanations by approximating the decision boundary of the baseline model trained to predict $\mathbf{Y}$ from $\mathbf{C}$ features only. CEN is trained to construct explanations given $\mathbf{C}$ and then make predictions by applying explanations to $\mathbf{X}$. The predictive performance of the produced explanations on noisy features is given on Figure 4a. Since baselines take only $\mathbf{C}$ as inputs, their performance stays the same and, regardless of the noise level, LIME "successfully" overfits explanations—it is able to almost perfectly approximate the decision boundary of the baselines using very noisy features. On the other hand, performance of CEN gets worse with the increasing noise level indicating that model fails to learn when the selected interpretable representation is of low quality.

**The effect of feature selection.** Here, we use the same setup, but instead of injecting noise into $\mathbf{X}$, we construct $\mathbf{X}$ by randomly subsampling a set of dimensions. Figure 4b demonstrates the result. While performance of CENs degrades proportionally to the size of $\mathbf{X}$, we see that, again, LIME is able to fit explanations to the decision boundary of the original models despite the loss of information.

---

[6] We use Gaussian noise with zero mean and select variance for each signal-to-noise ratio level appropriately.

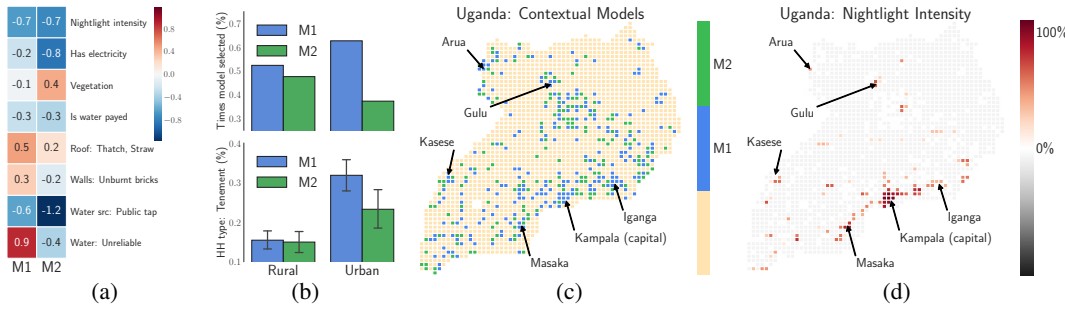

Figure 5: Qualitative results for the Satellite dataset: (a) Weights given to a subset of features by the two models (M1 and M2) discovered by CEN. (b) How frequently M1 and M2 are selected for areas marked rural or urban (top) and the average proportion of Tenement-type households in an urban/rural area for which M1 or M2 was selected. (c) M1 and M2 models selected for different areas on the Uganda map. M1 tends to be selected for more urbanized areas while M2 is picked for the rest. (d) Nightlight intensity of different areas of Uganda.

These two experiments indicate a major drawback of explaining predictions *post-hoc*: when constructed on poor, noisy, or incomplete features, such explanations can overfit the decision boundary of a predictor and are likely to be misleading. For example, predictions of a perfectly valid model might end up getting absurd explanations which is unacceptable from the decision support point of view.

### 4.1.3 QUALITATIVE ANALYSIS

Here, we focus on the poverty prediction task to analyze CEN-generated explanations qualitatively. Detailed discussion of qualitative results and visualization of the learned explanations for MNIST and IMDB datasets are given in Appendix F.2.

After training CEN with a dictionary of size 32, we discover that the encoder tends to sharply select one of the two explanations (M1 and M2) for different household clusters in Uganda (see Figure 5a, also Figure 13a in appendix). In the survey data, each household cluster is marked as either urban or rural. We notice that, conditional on a satellite image, CEN tends to pick M1 for urban areas and M2 for rural (Figure 5b). Notice that explanations weigh different categorical features, such as reliability of the water source or the proportion of houses with walls made of unburnt brick, quite differently. When visualized on the map, we see that CEN selects M1 more frequently around the major city areas, which also correlates with high nightlight intensity in those areas (Figures 5c,5d). High performance of the model makes us confident in the produced explanations (contrary to LIME as discussed in the previous section) and allows us to draw conclusions about what causes the model to classify certain households in different neighborhoods as poor.

### 4.2 SURVIVAL ANALYSIS AND STRUCTURED EXPLANATIONS

Finally, we apply CENs to survival analysis and showcase how to use our networks with structured explanations. In survival analysis, the goal is to learn a predictor for the time of occurrence of an event (in this case, the death of a patient) as well as be able to assess the risk (or hazard) of the occurrence. The classical models for this task are the Aalen's additive model (Aalen, 1989) and the Cox proportional hazard model (Cox, 1972), which linearly regress attributes of a particular patient, $\mathbf{X}$, to the hazard function. Yu et al. (2011) have shown that survival analysis can be formulated as a structured prediction problem and solved using a CRF variant. Here, we propose to use CENs with deep nets as encoders and CRF-structured explanations (as described in Section 3.3). More details on the architectures of our models, the baselines, as well as more background on survival analysis are provided in Appendix E.

**Datasets.** We use two publicly available datasets for survival analysis of of the intense care unit (ICU) patients: (a) SUPPORT2[7], and (b) data from the PhysioNet 2012 challenge[8]. The data was preprocessed and used as follows:

---

[7]http://biostat.mc.vanderbilt.edu/wiki/Main/DataSets.
[8]https://physionet.org/challenge/2012/.

Table 2: Performance of the classical Cox and Aalen models, CRF-based models, and CENs that use LSTM or MLP for context embedding and CRF for explanations. The numbers are averages from 5-fold cross-validation; the std. are on the order of the least significant digit. @K denotes the temporal quantile, i.e., the time point such that K% of the patients in the data have died or were censored before that point.

| | SUPPORT2 | | | | PhysioNet Challenge 2012 | | | | |
|---|---|---|---|---|---|---|---|---|---|
| Model | Acc@25 | Acc@50 | Acc@75 | RAE | Model | Acc@25 | Acc@50 | Acc@75 | RAE |
| Cox | 84.1 | 73.7 | 47.6 | 0.90 | Cox | 93.0 | 69.6 | 49.1 | 0.24 |
| Aalen | 87.1 | 66.2 | 45.8 | 0.98 | Aalen | 93.3 | 78.7 | 57.1 | 0.31 |
| CRF | 84.4 | 89.3 | 79.2 | 0.59 | CRF | 93.2 | 85.1 | 65.6 | 0.14 |
| MLP-CRF | **87.7** | 89.6 | 80.1 | 0.62 | LSTM-CRF | 93.9 | 86.3 | 68.1 | **0.11** |
| MLP-CEN | 85.5 | **90.8** | **81.9** | **0.56** | LSTM-CEN | **94.8** | **87.5** | **70.1** | **0.09** |

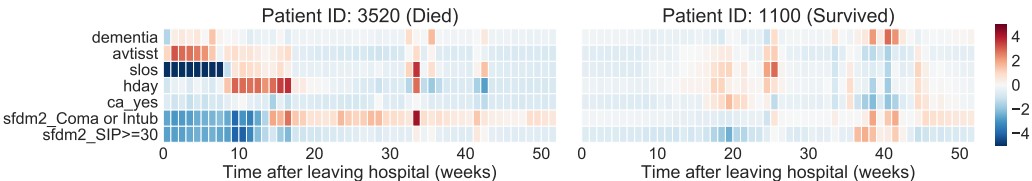

Figure 6: Weights of the CEN-generated CRF explanations for two patients from SUPPORT2 dataset for a set of the most influential features: `dementia` (comorbidity), `avtisst` (avg. TISS, days 3-25), `slos` (days from study entry to discharge), `hday` (day in hospital at study admit), `ca_yes` (the patient had cancer), `sfdm2_Coma or Intub` (intubated or in coma at month 2), `sfdm2_SIP` (sickness impact profile score at month 2). Higher weight values correspond to higher feature contributions to the risk of death after a given time.

- `SUPPORT2`: The data had 9105 patient records and 73 variables. We selected 50 variables for both **C** and **X** features (see Appendix F). Categorical features (such as `race` or `sex`) were one-hot encoded. The values of all features were non-negative, and we filled the missing values with -1. For CRF-based predictors, the survival timeline was capped at 3 years and converted into 156 discrete intervals of 7 days each. We used 7105 patient records for training, 1000 for validation, and 1000 for testing.

- `PhysioNet`: The data had 4000 patient records, each represented by a 48-hour irregularly sampled 37-dimensional time-series of different measurements taken during the patient's stay at the ICU. We resampled and mean-aggregated the time-series at 30 min frequency. This resulted in a large number of missing values that we filled with 0. The resampled time-series were used as the context, **C**, while for the attributes, **X**, we took the values of the last available measurement for each variable in the series. For CRF-based predictors, the survival timeline was capped at 60 days and converted into 60 discrete intervals.

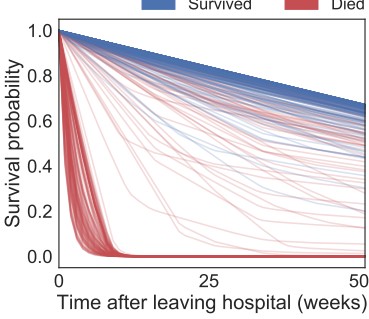

Figure 7: CEN-predicted survival curves for 500 random patients from SUP-PORT2 test set. Color indicates death within 1 year after leaving the hospital.

**Models.** For baselines, we use the classical Aalen and Cox models and the CRF from (Yu et al., 2011), where all used **X** as inputs. Next, we combine CRFs with neural encoders in two ways:

(i) We apply CRFs to the outputs from the neural encoders (the models denoted MLP-CRF and LSTM-CRF, all trainable end-to-end). Similar models have been show very successful in the natural language applications (Collobert et al., 2011). Note that parameters of the CRF layer assign weights to the latent features and are no longer interpretable in terms of the attributes of interest.

(ii) We use CENs with CRF-based explanations, that process the context variables, **C**, using the same neural networks as in (i) and output parameters for CRFs that act on the attributes, **X**.

Details on the architectures are given in Appendix F.3.

**Metrics.** Following Yu et al. (2011), we use two metrics specific to survival analysis: (a) accuracy of correctly predicting survival of a patient at times that correspond to 25%, 50%, and 75% population-level temporal quantiles (i.e., time points such that the corresponding % of the patients in the data were discharged from the study due to censorship or death) and (b) the relative absolute error (RAE) between the predicted and actual time of death for non-censored patients.

**Quantitative results.** The results for all models are given in Table 2. Our implementation of the CRF baseline reproduces (and even slightly improves) the performance reported by Yu et al. (2011). MLP-CRF and LSTM-CRF improve upon plain CRFs but, as we noted, can no longer be interpreted in terms of the original variables. On the other hand, CENs outperform neural CRF models on certain metrics (and closely match on the others) while providing explanations for risk prediction for each patient at each point in time.

**Qualitative results.** To inspect predictions of CENs qualitatively, for any given patient, we can visualize the weights assigned by the corresponding explanation to the respective attributes. Figure 6 explanation weights for a subset of the most influential features for two patients from SUPPORT2 dataset who were predicted as survivor and non-survivor. These explanations allow us to better understand patient-specific temporal dynamics of the contributing factors to the survival rates predicted by the model (Figure 7).

## 5 CONCLUSION

In this paper, we have introduced contextual explanation networks (CENs)—a class of models that learn to predict by generating and leveraging intermediate context-specific explanations. We have formally defined CENs as a class of probabilistic models, considered a number of special cases (e.g., the mixture of experts), and derived learning and inference procedures within the encoder-decoder framework for simple and sequentially-structured outputs. We have shown that, while explanations generated by CENs are provably equivalent to those generated *post-hoc* under certain conditions, there are cases when *post-hoc* explanations are misleading. Such cases are hard to detect unless explanation is a part of the prediction process itself. Besides, learning to predict and to explain jointly turned out to have a number of benefits, including strong regularization, consistency, and ability to generate explanations with no computational overhead.

We would like to point out a few limitations of our approach and potential ways of addressing those in the future work. Firstly, while each prediction made by CEN comes with an explanation, the process of conditioning on the context is still uninterpretable. Ideas similar to context selection (Liu et al., 2017) or rationale generation (Lei et al., 2016) may help improve interpretability of the conditioning. Secondly, the space of explanations considered in this work assumes the same graphical structure and parameterization for all explanations and uses a simple sparse dictionary constraint. This might be limiting, and one could imagine using a more hierarchically structured space of explanations instead, bringing to bear amortized inference techniques (Rudolph et al., 2017). Nonetheless, we believe that the proposed class of models is useful not only for improving prediction capabilities, but also for model diagnostics, pattern discovery, and general data analysis, especially when machine learning is used for decision support in high-stakes applications.

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

# A  ANALYSIS OF THE ASSUMPTIONS MADE BY CEN

As described in the main text, CENs represent the predictive distribution in the following form:

$$p(\mathbf{Y} \mid \mathbf{X}, \mathbf{C}) = \int p(\mathbf{Y} \mid \mathbf{X}, \boldsymbol{\theta})p(\boldsymbol{\theta} \mid \mathbf{C})d\boldsymbol{\theta}$$

and the assumed generative process behind the data is either:

(CEN) $\mathbf{Y} \sim p(\mathbf{Y} \mid \mathbf{X}, \boldsymbol{\theta})$, $\boldsymbol{\theta} \sim p(\boldsymbol{\theta} \mid \mathbf{C})$ for the purely discriminative setting.

(VCEN) $\mathbf{Y} \sim p(\mathbf{Y} \mid \mathbf{X}, \boldsymbol{\theta})$, $\boldsymbol{\theta} \sim p(\boldsymbol{\theta})$, $\mathbf{C} \sim p(\mathbf{C} \mid \boldsymbol{\theta})$ when we model the joint distribution of the explanations, $\boldsymbol{\theta}$, and contexts, $\mathbf{C}$, e.g., using encoder-decoder framework.

We would like to understand whether CEN, as defined above, can represent any conditional distribution, $p(\mathbf{Y} \mid \mathbf{X}, \mathbf{C})$, when the class of explanations is limited (e.g., to linear models), and, if not, what are the limitations?

Generally, CEN can be seen as a mixture of predictors. Such mixture models could be quite powerful as long as the mixing distribution, $p(\boldsymbol{\theta} \mid \mathbf{C})$, is rich enough. In fact, even a finite mixture exponential family regression models can approximate any smooth $d$-dimensional density at a rate $O(m^{-4/d})$ in the KL-distance (Jiang & Tanner, 1999). This result suggests that representing the predictive distribution with contextual mixtures should not limit the representational power of the model. The two caveats are:

(i) In practice, $p(\boldsymbol{\theta} \mid \mathbf{C})$ is limited, e.g., either deterministic encoding, a finite mixture, or a simple distribution parametrized by a deep network.

(ii) The classical setting of predictive mixtures does not separate inputs into two subsets, $(\mathbf{C}, \mathbf{X})$. We do this intentionally to produce hypotheses/explanations in terms of specific features that could be useful for interpretability or model diagnostics down the line. However, it could be the case that $\mathbf{X}$ contains only some limited information about $\mathbf{Y}$, which could limit the predictive power of the full model.

We leave the point (ii) to future work. To address (i), we consider $p(\boldsymbol{\theta} \mid \mathbf{C})$ that fully factorizes over the dimensions of $\boldsymbol{\theta}$: $p(\boldsymbol{\theta} \mid \mathbf{C}) = \prod_j p(\theta_j \mid \mathbf{C})$, and assume that hypotheses, $p(\mathbf{Y} \mid \mathbf{X}, \boldsymbol{\theta})$, factorize according to some underlying graph, $\mathcal{G}_{\mathbf{Y}} = (\mathcal{V}_{\mathbf{Y}}, \mathcal{E}_{\mathbf{Y}})$. The following proposition shows that in such case $p(\mathbf{Y} \mid \mathbf{X}, \mathbf{C})$ inherits the factorization properties of the hypothesis class.

**Proposition 1.** *Let $p(\boldsymbol{\theta} \mid \mathbf{C}) := \prod_j p(\theta_j \mid \mathbf{C})$ and let $p(\mathbf{Y} \mid \mathbf{X}, \boldsymbol{\theta})$ factorize according to some graph $\mathcal{G}_{\mathbf{Y}} = (\mathcal{V}_{\mathbf{Y}}, \mathcal{E}_{\mathbf{Y}})$. Then, $p(\mathbf{Y} \mid \mathbf{X}, \mathbf{C})$ defined by CEN with $p(\boldsymbol{\theta} \mid \mathbf{C})$ encoder and $p(\mathbf{Y} \mid \mathbf{X}, \boldsymbol{\theta})$ explanations also factorizes according to $\mathcal{G}$.*

**Proof.** Assume that $p(\mathbf{Y} \mid \mathbf{X}, \boldsymbol{\theta})$ factorizes as $\prod_{\boldsymbol{\alpha} \in \mathcal{V}_{\mathbf{Y}}} p(\mathbf{Y}_{\boldsymbol{\alpha}} \mid \mathbf{Y}_{\mathrm{MB}(\boldsymbol{\alpha})}, \mathbf{X}, \boldsymbol{\theta}_{\boldsymbol{\alpha}})$, where $\boldsymbol{\alpha}$ denotes subsets of the $\mathbf{Y}$ variables and $\mathrm{MB}(\boldsymbol{\alpha})$ stands for the corresponding Markov blankets. Using the definition of CEN, we have:

$$
\begin{aligned}
p(\mathbf{Y} \mid \mathbf{X}, \mathbf{C}) &= \int p(\mathbf{Y} \mid \mathbf{X}, \boldsymbol{\theta})p(\boldsymbol{\theta} \mid \mathbf{C})d\boldsymbol{\theta} &\text{(11)}\\
&= \int \prod_{\boldsymbol{\alpha} \in \mathcal{V}_{\mathbf{Y}}} p(\mathbf{Y}_{\boldsymbol{\alpha}} \mid \mathbf{Y}_{\mathrm{MB}(\boldsymbol{\alpha})}, \mathbf{X}, \boldsymbol{\theta}_{\boldsymbol{\alpha}}) \prod_j p(\theta_j \mid \mathbf{C})d\boldsymbol{\theta} &\text{(12)}\\
&= \prod_{\boldsymbol{\alpha} \in \mathcal{V}_{\mathbf{Y}}} \left[ \int p(\mathbf{Y}_{\boldsymbol{\alpha}} \mid \mathbf{Y}_{\mathrm{MB}(\boldsymbol{\alpha})}, \mathbf{X}, \boldsymbol{\theta}_{\boldsymbol{\alpha}}) \prod_{j \in \boldsymbol{\alpha}} p(\theta_j \mid \mathbf{C})d\boldsymbol{\theta}_{\boldsymbol{\alpha}} \right] &\text{(13)}\\
&= \prod_{\boldsymbol{\alpha} \in \mathcal{V}_{\mathbf{Y}}} p(\mathbf{Y}_{\boldsymbol{\alpha}} \mid \mathbf{Y}_{\mathrm{MB}(\boldsymbol{\alpha})}, \mathbf{X}, \mathbf{C}), &\text{(14)}
\end{aligned}
$$

$\square$

**Remark 1.** *All the encoding distributions, $p(\boldsymbol{\theta} \mid \mathbf{C})$, considered in the main text of the paper, including delta functions, their mixtures, and encoders parametrized by neural nets fully factorize over the dimensions of $\boldsymbol{\theta}$.*

**Remark 2.** *The proposition has no implications for the case of scalar targets, $\mathbf{Y}$. However, in case of structured prediction, regardless of how good the context encoder is, CEN will assume the same set of independencies as given by the class of hypotheses, $p(\mathbf{Y} \mid \mathbf{X}, \boldsymbol{\theta})$.*

## B  APPROXIMATING THE DECISION BOUNDARY OF CEN

Ribeiro et al. (2016) proposed to construct approximations of the of the decision boundary of an arbitrary predictor, $f$, in the locality of a specified point, $\mathbf{x}$, by solving the following optimization problem:

$$\hat{g} = \operatorname*{argmin}_{g \in G} \mathcal{L}(f, g, \pi_{\mathbf{x}}) + \Omega(g), \tag{15}$$

where $\mathcal{L}(f, g, \pi_{\mathbf{x}})$ measures the quality of $g$ as an approximation to $f$ in the neighborhood of $\mathbf{x}$ defined by $\pi_{\mathbf{x}}$ and $\Omega(g)$ is a regularizer that is usually used to ensure human-interpretability of the selected local hypotheses (e.g., sparsity). Now, consider the case when $f$ is defined by a CEN, instead of $\mathbf{x}$ we have $(\mathbf{c}, \mathbf{x})$, and the class of approximations, $G$, coincides with the class of explanations, and hence can be represented by $\boldsymbol{\theta}$. In this setting, we can pose the same problem as:

$$\hat{\boldsymbol{\theta}} = \operatorname*{argmin}_{\boldsymbol{\theta}} \mathcal{L}(f, \boldsymbol{\theta}, \pi_{\mathbf{c}, \mathbf{x}}) + \Omega(\boldsymbol{\theta}) \tag{16}$$

Suppose that CEN produces $\boldsymbol{\theta}^{\star}$ explanation for the context $\mathbf{c}$ using a deterministic encoder, $\phi$. The question is whether and under which conditions $\hat{\boldsymbol{\theta}}$ can recover $\boldsymbol{\theta}^{\star}$. Theorem 1 answers the question in affirmative and provides a concentration result for the case when hypotheses are linear. Here, we prove Theorem 1 for a little more general class of log-linear explanations: $\operatorname{logit} p(Y = 1 \mid \mathbf{x}, \theta) = \mathbf{a}(\mathbf{x})^{\top} \boldsymbol{\theta}$, where $\mathbf{a}$ is a $C$-Lipschitz vector-valued function whose values have a zero-mean distribution when $(\mathbf{x}, \mathbf{c})$ are sampled from $\pi_{\mathbf{x}, \mathbf{c}}$[9]. For simplicity of the analysis, we consider binary classification and omit the regularization term, $\Omega(g)$. We define the loss function, $\mathcal{L}(f, \boldsymbol{\theta}, \pi_{\mathbf{x}, \mathbf{c}})$, as:

$$\mathcal{L} = \frac{1}{K} \sum_{k=1}^{K} \left( \operatorname{logit} p(Y = 1 \mid \mathbf{x}_k - \mathbf{x}, \mathbf{c}_k) - \operatorname{logit} p(Y = 1 \mid \mathbf{x}_k - \mathbf{x}, \boldsymbol{\theta}) \right)^2, \tag{17}$$

where $(\mathbf{x}_k, \mathbf{c}_k) \sim \pi_{\mathbf{x}, \mathbf{c}}$ and $\pi_{\mathbf{x}, \mathbf{c}} := \pi_{\mathbf{x}} \pi_{\mathbf{c}}$ is a distribution concentrated around $(\mathbf{x}, \mathbf{c})$. Without loss of generality, we also drop the bias terms in the linear models and assume that $\mathbf{a}(\mathbf{x}_k - \mathbf{x})$ are centered.

**Proof of Theorem 1.**  The optimization problem (16) reduces to the least squares linear regression:

$$\hat{\boldsymbol{\theta}} = \operatorname*{argmin}_{\boldsymbol{\theta}} \frac{1}{K} \sum_{k=1}^{K} \left( \operatorname{logit} p(Y = 1 \mid \mathbf{x}_k - \mathbf{x}, \mathbf{c}_k) - \mathbf{a}(\mathbf{x}_k - \mathbf{x})^{\top} \boldsymbol{\theta} \right)^2 \tag{18}$$

We consider deterministic encoding, $p(\boldsymbol{\theta} \mid \mathbf{c}) := \delta(\boldsymbol{\theta}, \phi(\mathbf{c}))$, and hence $\operatorname{logit} p(Y = 1 \mid \mathbf{x}_k - \mathbf{x}, \mathbf{c}_k)$ takes the following form:

$$\operatorname{logit} p(Y = 1 \mid \mathbf{x}_k - \mathbf{x}, \mathbf{c}_k) = \operatorname{logit} p(Y = 1 \mid \mathbf{x}_k - \mathbf{x}, \boldsymbol{\theta} = \phi(\mathbf{c}_k)) = \mathbf{a}(\mathbf{x}_k - \mathbf{x})^{\top} \phi(\mathbf{c}_k) \tag{19}$$

To simplify the notation, we denote $\mathbf{a}_k := \mathbf{a}(\mathbf{x}_k - \mathbf{x})$, $\boldsymbol{\phi}_k := \phi(\mathbf{c}_k)$, and $\boldsymbol{\phi} := \phi(\mathbf{c})$. The solution of (18) now can be written in a closed form:

$$\hat{\boldsymbol{\theta}} = \left[ \frac{1}{K} \sum_{k=1}^{K} \mathbf{a}_k \mathbf{a}_k^{\top} \right]^{+} \left[ \frac{1}{K} \sum_{k=1}^{K} \mathbf{a}_k \mathbf{a}_k^{\top} \boldsymbol{\phi}_k \right] \tag{20}$$

Note that $\hat{\boldsymbol{\theta}}$ is a random variable since $(\mathbf{x}_k, \mathbf{c}_k)$ are randomly generated from $\pi_{\mathbf{x}, \mathbf{c}}$. To further simplify the notation, denote $M := \frac{1}{K} \sum_{k=1}^{K} \mathbf{a}_k \mathbf{a}_k^{\top}$. To get a concentration bound on $\|\hat{\boldsymbol{\theta}} - \boldsymbol{\theta}^{\star}\|$, we will use the continuity of $\phi(\cdot)$ and $\mathbf{a}(\cdot)$, concentration properties of $\pi_{\mathbf{x}, \mathbf{c}}$ around $(\mathbf{x}, \mathbf{c})$, and some elementary results from random matrix theory. To be more concrete, since we assumed that $\pi_{\mathbf{x}, \mathbf{c}}$ factorizes, we further let $\pi_{\mathbf{x}}$ and $\pi_{\mathbf{c}}$ concentrate such that $p_{\pi_{\mathbf{x}}}(\|\mathbf{x}' - \mathbf{x}\| > t) < \varepsilon_{\mathbf{x}}(t)$ and $p_{\pi_{\mathbf{c}}}(\|\mathbf{c}' - \mathbf{c}\| > t) < \varepsilon_{\mathbf{c}}(t)$, respectively, where $\varepsilon_{\mathbf{x}}(t)$ and $\varepsilon_{\mathbf{c}}(t)$ both go to 0 as $t \to \infty$, potentially at different rates.

---

[9]In case of logistic regression, $\mathbf{a}(\mathbf{x}) = [1, x_1, \dots, x_d]^{\top}$.

First, we have the following bound from the convexity of the norm:

$$p(\|\hat{\boldsymbol{\theta}} - \boldsymbol{\theta}^\star\| > t) \quad = \quad p(\left\| \frac{1}{K} \sum_{k=1}^K \left[ M^+ \mathbf{a}_k \mathbf{a}_k^\top (\boldsymbol{\phi}_k - \boldsymbol{\phi}) \right] \right\| > t) \tag{21}$$

$$\leq \quad p(\frac{1}{K} \sum_{k=1}^K \left\| M^+ \mathbf{a}_k \mathbf{a}_k^\top (\boldsymbol{\phi}_k - \boldsymbol{\phi}) \right\| > t) \tag{22}$$

By making use of the inequality $\|Ax\| \leq \|A\|\|x\|$, where $\|A\|$ denotes the spectral norm of the matrix $A$, the $L$-Lipschitz property of $\phi(\mathbf{c})$, the $C$-Lipschitz property of $\mathbf{a}(\mathbf{x})$, and the concentration of $\mathbf{x}_k$ around $\mathbf{x}$, we have

$$p(\|\hat{\boldsymbol{\theta}} - \boldsymbol{\theta}^\star\| > t) \quad \leq \quad p(L \frac{1}{K} \sum_{k=1}^K \left\| M^+ \mathbf{a}_k \mathbf{a}_k^\top \right\| \|\mathbf{c}_k - \mathbf{c}\| > t) \tag{23}$$

$$\leq \quad p(CL \left\| M^+ \right\| \frac{1}{K} \sum_{k=1}^K \left\| \mathbf{a}_k \mathbf{a}_k^\top \right\| \|\mathbf{c}_k - \mathbf{c}\| > t) \tag{24}$$

$$\leq \quad p(\frac{CL}{\lambda_{\min}(M)} \frac{1}{K} \sum_{k=1}^K \|\mathbf{x}_k - \mathbf{x}\| \|\mathbf{c}_k - \mathbf{c}\| > t) \tag{25}$$

$$\leq \quad p(\frac{CL\tau^2}{\lambda_{\min}(M)} > t) + p(\|\mathbf{x}_k - \mathbf{x}\| \|\mathbf{c}_k - \mathbf{c}\| > \tau^2) \tag{26}$$

$$\leq \quad p(\lambda_{\min}\left(M/(C\tau)^2\right) < \frac{L}{C^2 t}) + \varepsilon_\mathbf{x}(\tau) + \varepsilon_\mathbf{c}(\tau) \tag{27}$$

Note that we used the fact that the spectral norm of a rank-1 matrix, $\mathbf{a}(\mathbf{x}_k)\mathbf{a}(\mathbf{x}_k)^\top$, is simply the norm of $\mathbf{a}(\mathbf{x}_k)$, and the spectral norm of the pseudo-inverse of a matrix is equal to the inverse of the least non-zero singular value of the original matrix: $\|M^+\| \leq \lambda_{\max}(M^+) = \lambda_{\min}^{-1}(M)$.

Finally, we need a concentration bound on $\lambda_{\min}\left(M/(C\tau)^2\right)$ to complete the proof. Note that $\frac{M}{C^2 \tau^2} = \frac{1}{K} \sum_{k=1}^K \left(\frac{\mathbf{a}_k}{C\tau}\right)\left(\frac{\mathbf{a}_k}{C\tau}\right)^\top$, where the norm of $\left(\frac{\mathbf{a}_k}{C\tau}\right)$ is bounded by 1. If we denote $\mu_{\min}(C\tau)$ the minimal eigenvalue of $\mathsf{Cov}\left[\frac{\mathbf{a}_k}{C\tau}\right]$, we can write the matrix Chernoff inequality (Tropp, 2012) as follows:

$$p(\lambda_{\min}\left(M/(C\tau)^2\right) < \alpha) \leq d\exp\left\{-KD(\alpha\|\mu_{\min}(C\tau))\right\}, \quad \alpha \in [0, \mu_{\min}(C\tau)],$$

where $d$ is the dimension of $\mathbf{a}_k$, $\alpha := \frac{L}{C^2 t}$, and $D(a\|b)$ denotes the binary information divergence:

$$D(a\|b) = a\log\left(\frac{a}{b}\right) + (1 - a)\log\left(\frac{1 - a}{1 - b}\right).$$

The final concentration bound has the following form:

$$p(\|\hat{\boldsymbol{\theta}} - \boldsymbol{\theta}^\star\| > t) \leq d\exp\left\{-KD\left(\frac{L}{C^2 t}\|\mu_{\min}(C\tau)\right)\right\} + \varepsilon_\mathbf{x}(\tau) + \varepsilon_\mathbf{c}(\tau) \tag{28}$$

We see that as $\tau \to \infty$ and $t \to \infty$ all terms on the right hand side vanish, and hence $\hat{\boldsymbol{\theta}}$ concentrates around $\boldsymbol{\theta}^\star$. Note that as long as $\mu_{\min}(C\tau)$ is far from 0, the first term can be made negligibly small by sampling more points around $(\mathbf{x}, \mathbf{c})$. Finally, we set $\tau \equiv t$ and denote the right hand side by $\delta_{K,L,C}(t)$ that goes to 0 as $t \to \infty$ to recover the statement of the original theorem. $\square$

**Remark 3.** *We have shown that $\hat{\boldsymbol{\theta}}$ concentrates around $\boldsymbol{\theta}^\star$ under mild conditions. With more assumptions on the sampling distribution, $\pi_{\mathbf{x},\mathbf{c}}$, (e.g., sub-gaussian) one could derive precise convergence rates. Note that we are in total control of any assumptions we put on $\pi_{\mathbf{x},\mathbf{c}}$ since precisely that distribution is used for sampling. This is a major difference between the local approximation setup here and the setup of linear regression with random design; in the latter case, we have no control over the distribution of the design matrix, and any assumptions we make could potentially be unrealistic.*

**Remark 4.** *Note that concentration analysis of a more general case when the loss $\mathcal{L}$ is a general convex function and $\Omega(g)$ is a decomposable regularizer could be done by using results from the M-estimation theory (Negahban et al., 2009), but would be much more involved and unnecessary for our purposes.*

## C   Learning and Inference in the Contextual Mixture of Experts

As noted in the main text, to make a prediction, MoE uses each of the $K$ experts where the predictive distribution is computed as follows:

$$p_{\mathbf{w},\mathbf{D}}(\mathbf{Y} \mid \mathbf{X}, \mathbf{C}) = \sum_{k=1}^{K} p_{\mathbf{w}}(k|\mathbf{C})p(\mathbf{Y} \mid \mathbf{X}, \boldsymbol{\theta}_k). \tag{29}$$

Since each expert contributes to the predictive probability, we can explain a prediction, $\hat{\mathbf{y}}$, for the instance $(\mathbf{x}, \mathbf{c})$ in terms of the posterior weights assigned to each expert model:

$$p_{\mathbf{w}}(k \mid \hat{\mathbf{y}}, \mathbf{x}, \mathbf{c}) = \frac{p_{\mathbf{w}}(\hat{\mathbf{y}}, k \mid \mathbf{x}, \mathbf{c})}{p_{\mathbf{w}}(\hat{\mathbf{y}} \mid \mathbf{x}, \mathbf{c})} = \frac{p(\hat{\mathbf{y}} \mid \mathbf{x}, \boldsymbol{\theta}_k)p_{\mathbf{w}}(k \mid \mathbf{c})}{\sum_{k=1}^{K} p_{\mathbf{w}}(k|\mathbf{c})p(\hat{\mathbf{y}} \mid \mathbf{x}, \boldsymbol{\theta}_k)} \tag{30}$$

If the $p(k \mid \hat{\mathbf{y}}, \mathbf{x}, \mathbf{c})$ assigns very high weight to a single expert, we can treat that expert model as an explanation. Note that however, in general, this may not be the case and posterior weights could be quite spread out (especially, if the number of experts is small and the class of expert models, $p(\mathbf{Y} \mid \mathbf{X}, \boldsymbol{\theta})$, is too simple and limited). Therefore, there may not exist an equivalent local explanation in the class of expert models that would faithfully approximate the decision boundary.

To learn contextual MoE, we can either directly optimize the conditional log-likelihood, which is non-convex yet tractable, or use expectation maximization (EM) procedure. For the latter, we write the log likelihood in the following form:

$$\log p(\mathbf{y} \mid \mathbf{x}, \mathbf{c}) = \sum_{k=1}^{K} q(k) \log p(\mathbf{y} \mid \mathbf{x}, \mathbf{c}) \tag{31}$$

$$= \sum_{k=1}^{K} q(k) \log \frac{p(\mathbf{y} \mid \mathbf{x}, \boldsymbol{\theta}_k)p_{\mathbf{w}}(k \mid \mathbf{c})q(k)}{p_{\mathbf{w}}(k \mid \mathbf{y}, \mathbf{x}, \mathbf{c})q(k)} \tag{32}$$

$$= \sum_{k=1}^{K} q(k) \log \frac{p(\mathbf{y} \mid \mathbf{x}, \boldsymbol{\theta}_k)p_{\mathbf{w}}(k \mid \mathbf{c})}{q(k)} + \mathrm{KL}\left(q(k)\| \, p_{\mathbf{w}}(k \mid \mathbf{y}, \mathbf{x}, \mathbf{c})\right) \tag{33}$$

At each iteration, we do two steps:

(E-step) Compute posteriors for each data instance, $q_i(k) = p_{\mathbf{w}}(k \mid \mathbf{y}_i, \mathbf{x}_i, \mathbf{c}_i)$.

(M-step) Optimize $Q(\mathbf{w}) = \sum_{k=1}^{K} q(k) \log p(\mathbf{y} \mid \mathbf{x}, \boldsymbol{\theta}_k)p_{\mathbf{w}}(k \mid \mathbf{c})$.

It is well known that this iterative procedure is guaranteed to converge to a local optimum.

## D   Contextual Variational Autoencoders

We can express the evidence for contextual variational autoencoders as follows:

$$\log p(\mathbf{Y}, \mathbf{C} \mid \mathbf{X})$$

$$= \int q_{\mathbf{w}}(\boldsymbol{\theta} \mid \mathbf{C}) \log p(\mathbf{Y}, \mathbf{C} \mid \mathbf{X})d\boldsymbol{\theta}$$

$$= \int q_{\mathbf{w}}(\boldsymbol{\theta} \mid \mathbf{C}) \log \frac{p(\mathbf{Y} \mid \mathbf{X}, \boldsymbol{\theta})p_{\mathbf{u}}(\mathbf{C} \mid \boldsymbol{\theta})p(\boldsymbol{\theta})}{p_{\mathbf{u}}(\boldsymbol{\theta} \mid \mathbf{Y}, \mathbf{X}, \mathbf{C})}d\boldsymbol{\theta}$$

$$= \int q_{\mathbf{w}}(\boldsymbol{\theta} \mid \mathbf{C}) \log \frac{p(\mathbf{Y} \mid \mathbf{X}, \boldsymbol{\theta})p_{\mathbf{u}}(\mathbf{C} \mid \boldsymbol{\theta})p(\boldsymbol{\theta})}{q_{\mathbf{w}}(\boldsymbol{\theta} \mid \mathbf{C})}d\boldsymbol{\theta} + \mathrm{KL}\left(q_{\mathbf{w}}(\boldsymbol{\theta} \mid \mathbf{C})\| \, p_{\mathbf{u}}(\boldsymbol{\theta} \mid \mathbf{Y}, \mathbf{X}, \mathbf{C})\right)$$

$$\geq \mathcal{L}(\mathbf{w}, \mathbf{u}; \mathbf{Y}, \mathbf{X}, \mathbf{C}), \tag{34}$$

where $\mathcal{L}(\mathbf{w}, \mathbf{u}; \mathbf{Y}, \mathbf{X}, \mathbf{C})$ is the evidence lower bound (ELBO):

$$\mathcal{L}(\mathbf{w}, \mathbf{u}; \mathbf{Y}, \mathbf{X}, \mathbf{C})$$

$$= \mathbb{E}_{q_{\mathbf{w}}}\left[\log \frac{p(\mathbf{Y} \mid \mathbf{X}, \boldsymbol{\theta})p_{\mathbf{u}}(\mathbf{C} \mid \boldsymbol{\theta})p(\boldsymbol{\theta})}{q_{\mathbf{w}}(\boldsymbol{\theta} \mid \mathbf{C})}\right] \tag{35}$$

$$= \mathbb{E}_{q_{\mathbf{w}}}\left[\log p(\mathbf{Y} \mid \mathbf{X}, \boldsymbol{\theta})\right] + \mathbb{E}_{q_{\mathbf{w}}}\left[\log p_{\mathbf{u}}(\mathbf{C} \mid \boldsymbol{\theta})\right] - \mathrm{KL}\left(q(\boldsymbol{\theta} \mid \mathbf{C})\| \, p(\boldsymbol{\theta})\right)$$

We notice that the ELBO consists of three terms:

(1) the expected conditional likelihood of the explanation, $\mathbb{E}_{q_{\mathbf{w}}} \left[ \log p(\mathbf{Y} \mid \mathbf{X}, \boldsymbol{\theta}) \right]$,

(2) the expected context reconstruction error, $\mathbb{E}_{q_{\mathbf{w}}} \left[ \log p_{\mathbf{u}}(\mathbf{C} \mid \boldsymbol{\theta}) \right]$, and

(3) the KL-based regularization term, $-\mathrm{KL} \left( q(\boldsymbol{\theta} \mid \mathbf{C}) \| p(\boldsymbol{\theta}) \right)$.

We can optimize the ELBO using first-order methods by estimating the gradients via Monte Carlo sampling with reparametrization. When the encoder has a classical form of a Gaussian distribution (or any other location-scale type of distribution), $q_{\mathbf{w}}(\boldsymbol{\theta} \mid \mathbf{C}) = \mathcal{N} \left( \boldsymbol{\theta}; \boldsymbol{\mu}_{\mathbf{w}}(\mathbf{C}), \mathrm{diag} \left( \boldsymbol{\sigma}_{\mathbf{w}}(\mathbf{C}) \right) \right)$, reparametrization of the samples is straightforward (Kingma & Welling, 2013).

In our experiments, we mainly consider encoders that output probability distributions over a simplex spanned by a dictionary, $\mathbf{D}$, which turned out to have better performance and faster convergence. In particular, sampling from the encoder is as follows:

$$
\begin{aligned}
&\mathbf{z} \sim \mathcal{N} \left( \boldsymbol{\theta}; \boldsymbol{\mu}_{\mathbf{w}}(\mathbf{C}), \mathrm{diag} \left( \boldsymbol{\sigma}_{\mathbf{w}}(\mathbf{C}) \right) \right), \\
&\gamma_0 = \frac{1}{1 + \sum_{j=1}^{K} e^{z_j}}, \; \gamma_i = \frac{e^{z_i}}{1 + \sum_{j=1}^{K} e^{z_j}}, \; i = 1, \ldots, K, \\
&\boldsymbol{\theta} = \mathbf{D} \cdot \boldsymbol{\gamma}
\end{aligned}
\tag{36}
$$

The samples, $\boldsymbol{\theta}$, will be logistic normal distributed and are easy to be re-parametrized. For prior, we use the Dirichlet distribution over $\boldsymbol{\gamma}$ with the parameter vector $\boldsymbol{\alpha}$. In that case, the stochastic estimate of the KL-based regularization term has the following form:

$$
-\mathrm{KL} \left( q(\boldsymbol{\gamma} \mid \mathbf{C}) \| p(\boldsymbol{\gamma}) \right) \simeq \frac{1}{L} \sum_{l=1}^{L} \log \mathcal{N} \left( \log \left( \frac{\boldsymbol{\gamma}_{-0}^{(l)}}{\gamma_0^{(l)}} \right); \boldsymbol{\mu}_{\mathbf{w}}(\mathbf{C}), \mathrm{diag} \left( \boldsymbol{\sigma}_{\mathbf{w}}(\mathbf{C}) \right) \right) - \boldsymbol{\alpha}^\top \log(\boldsymbol{\gamma}^{(l)}),
\tag{37}
$$

where $\boldsymbol{\gamma}_{-0}^{(l)}$ is a parameter vector without the first element, and $l$ indexes samples taken from the encoder, $p(\boldsymbol{\gamma} \mid \mathbf{C})$. In practice, we use $L = 1$.

## E  SURVIVAL ANALYSIS AND CONTEXTUAL CONDITIONAL RANDOM FIELDS

We provide some general background on survival analysis, the classical Aalen additive hazard (Aalen, 1989) and Cox proportional hazard (Cox, 1972) models, derive the structured prediction approach (Yu et al., 2011), and describe CENs with CRF-based explanations used in our experiments in detail.

In survival time prediction, our goal is to estimate the occurrence time of an event in the future (e.g., death of a patient, earthquake, hard drive failure, customer turnover, etc.). The unique aspect of the survival data is that there is always a fraction of points for which the event time has not been observed (such data instances are called *censored*). The common approach is to model the survival time, $T$, either for a population (i.e., average survival time) or for each instance. In particular, we can introduce the *survival function*, $S(t) := p(T \geq t)$, which gives the probability of the event *not* happening at least up to time $t$ (e.g., patient survived up to time $t$). The derivative of the survival function is called the *hazard function*, $\lambda(t)$, which is the instantaneous rate of failure:

$$
S(t) := - \int_0^t \lambda(\tau) d\tau
\tag{38}
$$

This allows us to model survival on a population level. Now, proportional hazard models assume that $\lambda$ is also a function of the available features of a given instance, i.e., $\lambda(t; \mathbf{x})$. Cox's proportional hazard model assumes $\lambda_{\boldsymbol{\theta}}(t; \mathbf{x}) := \lambda_0(t) \exp(\mathbf{x}^\top \boldsymbol{\theta})$. Aalen's model is a time-varying extension and assumes that $\lambda_{\boldsymbol{\theta}}(t; \mathbf{x}) := \mathbf{x}^\top \boldsymbol{\theta}(t)$, where $\boldsymbol{\theta}(t)$ is a function of time.

Survival analysis is a regression problem as it originally works with continuous time. The time can be discretized (e.g., into days, months, etc.), and hence we can approach survival time prediction as a multi-task classification problem (Efron, 1988). Yu et al. (2011) went one step further, noted that the output space is structure in a particular way, and proposed a model called *sequence of dependent regressors*, which is in essence a conditional random field with a particular structure of the pairwise potentials between the labels. In particular, as we described in Section 3.3, the targets are sequences

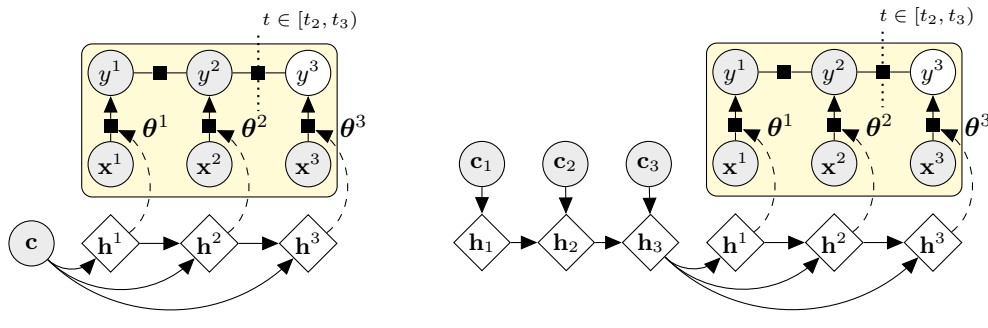

(a) Architecture used for SUPPORT2        (b) Architecture used for PhysioNet

Figure 8: CEN architectures used in our survival analysis experiments. Context encoders were time-distributed single hidden layer MLP (a) and LSTM (b) that produced inputs for another LSTM over the output time intervals (denoted with $\mathbf{h}^1$, $\mathbf{h}^2$, $\mathbf{h}^3$ hidden states respectively). Each hidden state of the output LSTM was used to generate the corresponding $\boldsymbol{\theta}^t$ that were further used to construct the log-likelihood for CRF.

of binary random variables, $\mathbf{Y} := (y^1, \ldots, y^m)$, that encode occurrence of an event as follows: for an event that occurred at time $t \in [t_i, t_{i+1})$, then $y^j = 0$, $\forall j \leq i$ and $y^k = 1$, $\forall k > i$. Note that only $m + 1$ sequences are valid, i.e., assigned non-zero probability by the model, which allows us to write the following linear model:

$$p(\mathbf{Y} = (y^1, \ldots, y^m) \mid \mathbf{x}, \boldsymbol{\Theta}) = \frac{\exp\left(\sum_{t=1}^m y^t \mathbf{x}^\top \boldsymbol{\theta}^t\right)}{\sum_{k=0}^m \exp\left(\sum_{t=k+1}^m \mathbf{x}^\top \boldsymbol{\theta}^t\right)} \tag{39}$$

To train the model, Yu et al. (2011) optimize the following objective:

$$\min_{\boldsymbol{\Theta}} C_1 \sum_{t=1}^m \|\theta^t\|^2 + C_2 \sum_{t=1}^{m-1} \|\theta^{t+1} - \theta^t\|^2 - \log \mathcal{L}(\mathbf{Y}, \mathbf{X}; \boldsymbol{\Theta}) \tag{40}$$

where the first two terms are regularization and the last term is the log-likelihood which as:

$$\mathcal{L}(\mathbf{Y}, \mathbf{X}; \boldsymbol{\Theta}) = \sum_{i \in \mathrm{NC}} p(T = t_i \mid \mathbf{x}_i, \boldsymbol{\Theta}) + \sum_{j \in \mathrm{C}} p(T > t_j \mid \mathbf{x}_j, \boldsymbol{\Theta}) \tag{41}$$

where NC is the set of non-censored instances (for which we know the outcome times, $t_i$) and C is the set of censored inputs (for which only know the censorship times, $t_j$). Expressions for the likelihoods of censored and non-censored inputs are the same as given in Section 3.3.

Finally, CENs additionally take the context variables, $\mathbf{C}$, as inputs and generate $\boldsymbol{\theta}^t$ for each time step using a recurrent encoder. In our experiments, we considered datasets where the context was represented by a vector or regularly sampled time series. Architectures for CENs used in our experiments are given in Figure 8. We used encoders suitable for the data type of the context variables available for each dataset. Each $\boldsymbol{\theta}^t$ was generated using a constrained deterministic encoder with a global dictionary, $\mathbf{D}$ of size 16. For details on parametrization of our architectures see tables in Appendix F.3.

Importantly, CEN-CRF architectures are trainable end-to-end (as all other CEN architectures considered in this paper), and we optimized the objective using stochastic gradient method. For each mini-batch, depending on which instances were censored and which were non-censored, we constructed the objective function accordingly (to implement this in TensorFlow we used masking and the standard control flow primitives for selecting between parts of the objective for censored and non-censored inputs).

## F    EXPERIMENTAL DETAILS

This section provides details on the experimental setups including architectures, training procedures, etc. Additionally, we provide and discuss qualitative results for CENs on MNIST and IMDB datasets.

## F.1  ADDITIONAL DETAILS ON THE DATASETS AND EXPERIMENT SETUPS

**MNIST.** We used the classical split of the dataset into 50k training, 10k validation, and 10k testing points. All models were trained for 100 epochs using the Adam optimizer with the learning rate of $10^{-3}$. No data augmentation was used in any of our experiments. HOG representations were computed using $3 \times 3$ blocks.

**CIFAR10.** For this set of experiments, we followed the setup given Zagoruyko (2015), reimplemented in Keras with TensorFlow backend. The input images were global contrast normalized (a.k.a. GCN whitened) while the rescaled image representations were simply standardized. Again, HOG representations were computed using $3 \times 3$ blocks. No data augmentation was used in our experiments.

**IMDB.** We considered the labeled part of the data only (50,000 reviews total). The data were split into 20,000 train, 5,000 validation, and 25,000 test points. The vocabulary was limited to All models were trained with the Adam optimizers with $10^{-2}$ learning rate. The models were initialized randomly; no pre-training or any other unsupervised/semi-supervised technique was used.

**Satellite.** As described in the main text, we used a pre-trained `VGG-16` network[10] to extract features from the satellite imagery. Further, we added one fully connected layer network with 128 hidden units used as the context encoder. For the VCEN model, we used dictionary-based encoding with Dirichlet prior and logistic normal distribution as the output of the inference network. For the decoder, we used an MLP of the same architecture as the encoder network. All models were trained with Adam optimizer with 0.05 learning rate. The results were obtained by 5-fold cross-validation.

**Medical data.** We have used minimal pre-processing of both SUPPORT2 and PhysioNet datasets limited to standardization and missing-value filling. We found that denoting missing values with negative entries $(-1)$ often led a slightly improved performance compared to any other NA-filling techniques. PhysioNet time series data was irregularly sampled across the time, so we had to resample temporal sequences at regular intervals of 30 minutes (consequently, this has created quite a few missing values for some of the measurements). All models were trained using Adam optimizer with $10^{-2}$ learning rate.

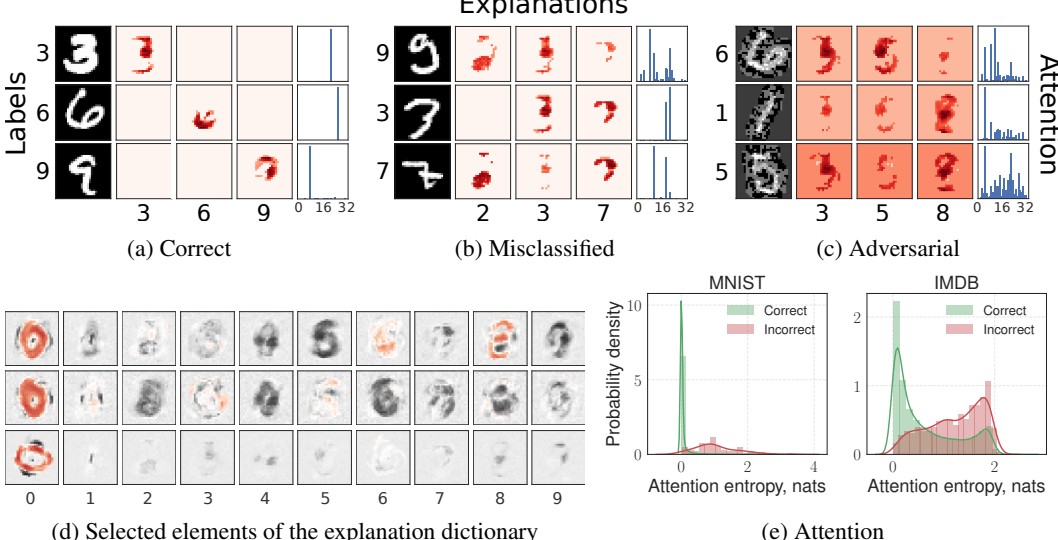

Figure 9: Explanations generated by CEN for the 3 top classes and the corresponding attention vectors for (a) correctly classified, (b) misclassified, and (c) adversarially constructed images. Adversarial examples were generated using the fast gradient sign method (FGSM) (Papernot et al., 2016). (d) Elements from the learned 32-element dictionary that correspond to different writing styles of 0 digits. (e) Histogram of the attention entropy for correctly and incorrectly classified test instances for CEN-`pxl` on MNIST and CEN-`tpc` on IMDB.

---

[10]The model was taken form `https://github.com/nealjean/predicting-poverty`.

### F.2 MORE ON QUALITATIVE ANALYSIS

Here, we discuss additional qualitative results obtained for CENs on MNIST and IMDB data.

### F.2.1 MNIST

Figures 9a, 9b, and 9c visualize explanations for predictions made by CEN-pxl on MNIST. The figures correspond to 3 cases where CEN (a) made a correct prediction, (b) made a mistake, and (c) was applied to an adversarial example (and made a mistake). Each chart consists of the following columns: true labels, input images, explanations for the top 3 classes (as given by the activation of the final softmax layer), and attention vectors used to select explanations from the global dictionary. A small subset of explanations from the dictionary is visualized in Figure 9d (the full dictionary is given in Figure 11), where each image is a weight vector used to construct the pre-activation for a particular class. Note that different elements of the dictionary capture different patterns in the data (in Figure 9d, different styles of writing the 0 digit) which CEN actually uses for prediction.

Also note that confident correct predictions (Figures 9a) are made by selecting a single explanation from the dictionary using a sharp attention vector. However, when the model makes a mistake, its attention is often dispersed (Figures 9b and 9c), i.e., there is uncertainty in which pattern it tries to use for prediction. Figure 9e further quantifies this phenomenon by plotting histogram of the attention entropy for all test examples which were correctly and incorrectly classified. While CENs are certainly not adversarial-proof, high entropy of the attention vectors is indicative of ambiguous or out-of-distribution examples which is helpful for model diagnostics.

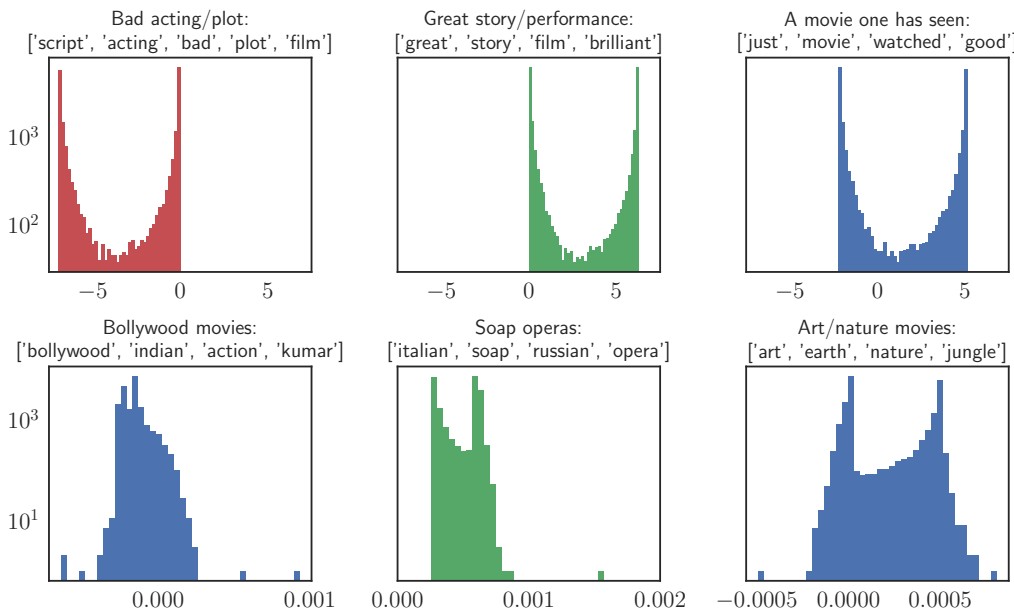

Figure 10: Histograms of test weights assigned by CEN to 6 topics: Acting- and plot-related topics (upper charts), genre topics (bottom charts). Note that acting-related topics are often bi-modal, i.e., contributing either positive, negative, or zero weight to the sentiment prediction in different contexts. Genre topics almost always have negligible contributions. This allows us to conclude that the learned model does not have any particular biases towards or against any a given genre.

### F.2.2 IMDB

Similar to MNIST, we train CEN-tpc with linear explanations in terms of topics on the IMDB dataset. Then, we generate explanations for each test example and visualize histograms of the weights assigned by the explanations to 6 selected topics in Figure 10. The 3 topics in the top row are acting- and plot-related (and intuitively have positive, negative, or neutral connotation), while the 3 topics in the bottom are related to particular genre of the movies.

Note that acting-related topics turn out to be bi-modal, i.e., contributing either positively, negatively, or neutrally to the sentiment prediction in different contexts. As expected intuitively, CEN assigns highly negative weight to the topic related to "bad acting/plot" and highly positive weight to "great story/performance" in most of the contexts (and treats those neutrally conditional on some of the reviews). Interestingly, genre-related topics almost always have a negligible contribution to the sentiment (i.e., get almost 0 weights assigned by explanations) which indicates that the learned model does not have any particular bias towards or against a given genre. Importantly, inspecting summary statistics of the explanations generated by CEN allows us to explore the biases that the model picks up from the data and actively uses for prediction[11].

Figure 12 visualizes the full dictionary of size 16 learned by CEN-tpc. Each column corresponds to a dictionary atom that represents a typical explanation pattern that CEN attends to before making a prediction. By inspecting the dictionary, we can find interesting patterns. For instance, atoms 5 and 11 assign inverse weights to topics [kid, child, disney, family] and [sexual, violence, nudity, sex]. Depending on the context of the review, CEN may use one of these patterns to predict the sentiment. Note that these two topics are negatively correlated across all dictionary elements, which again is quite intuitive.

### F.2.3 SATELLITE

We visualize the two explanations, M1 and M2, learned by CEN-att on the Satellite dataset in full in Figures 13a and provide additional correlation plots between the selected explanation and values of each survey variable in Figure 13b.

### F.3 MODEL ARCHITECTURES

Architectures of the model used in our experiments are summarized in Tables 3, 4, 5.

---

[11]If we wish to enforce or eliminate certain patterns from explanations (e.g., to ensure fairness), we may impose additional constraints on the dictionary. However, this is beyond the scope of this work.

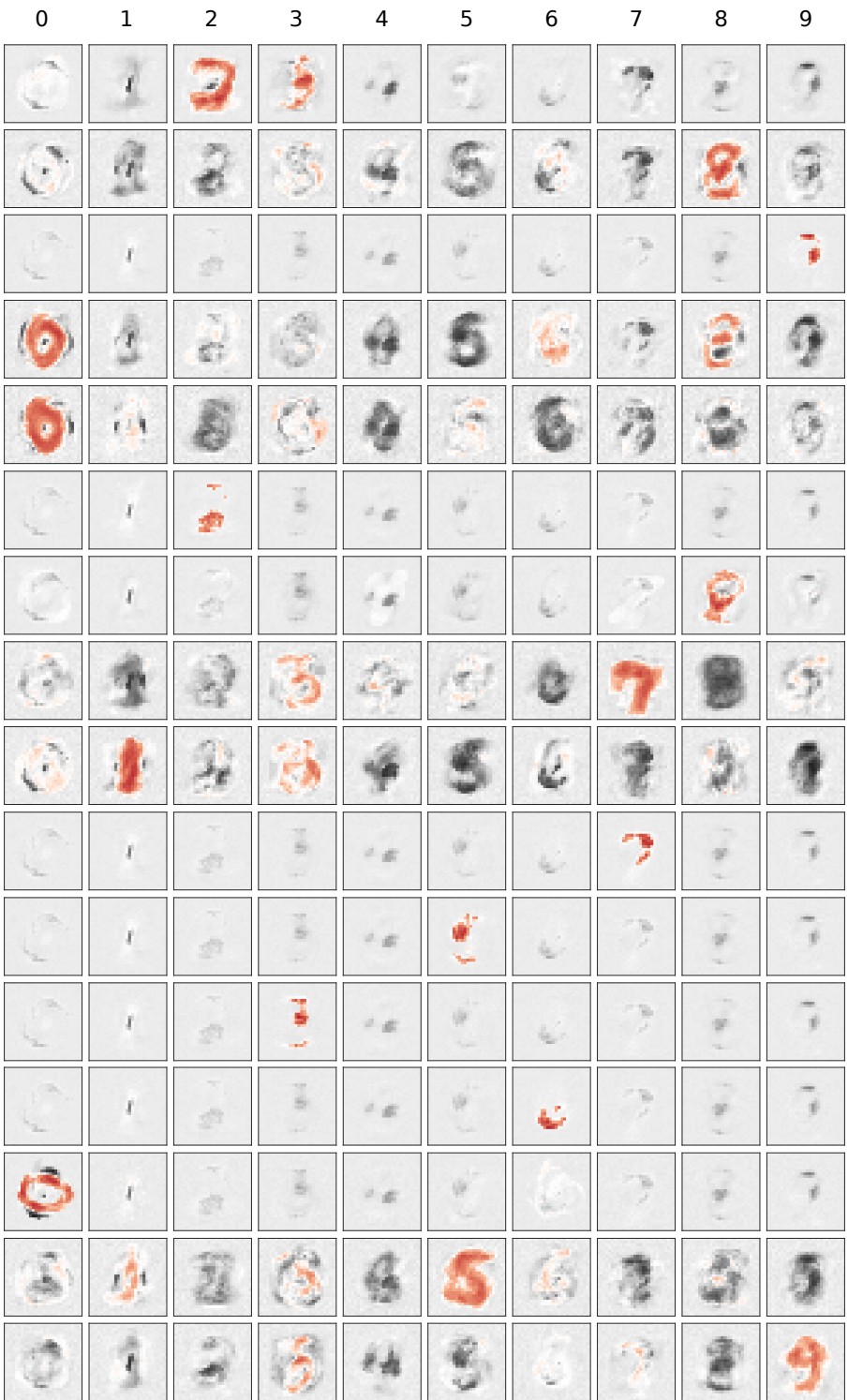

Figure 11: Visualization of the model dictionary learned by CEN on MNIST. Each row corresponds to a dictionary element, and each column corresponds to the weights of the model voting for each class of digits. Images visualize the weights of the models. Red corresponds to high positive values, dark gray to high negative values, and white to values that are close to 0.

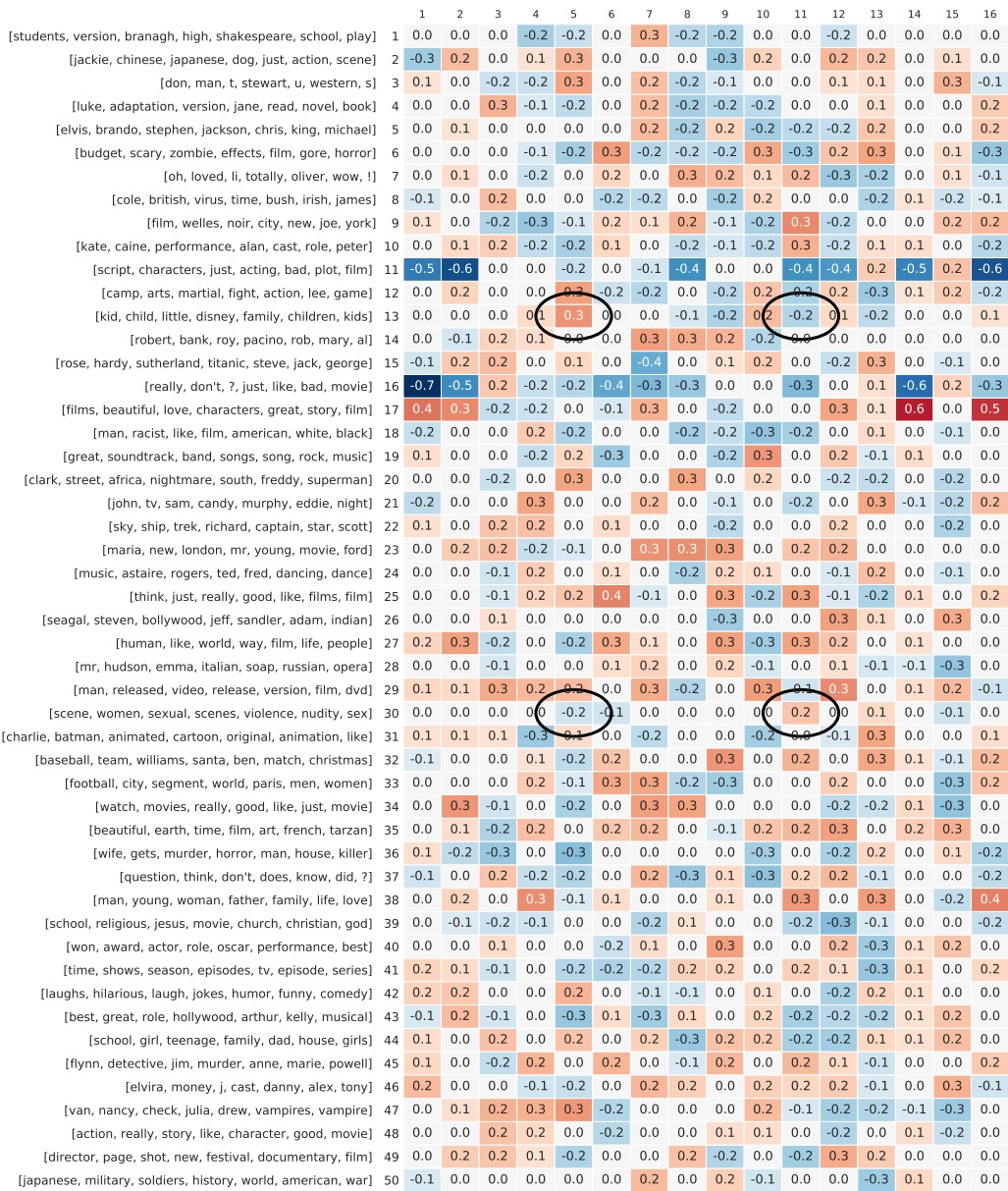

Figure 12: The full dictionary learned by CEN$_{\text{tpc}}$ model: rows correspond to topics and columns correspond to dictionary atoms. Very small values were thresholded for visualization clarity. Different atoms capture different prediction patterns; for example, atom 5 assigns a highly positive weight to the `[kid, child, disney, family]` topic and down-weighs `[sexual, violence, nudity, sex]`, while atom 11 acts in an opposite manner. Given the context of the review, CEN combines just a few atoms to make a prediction.

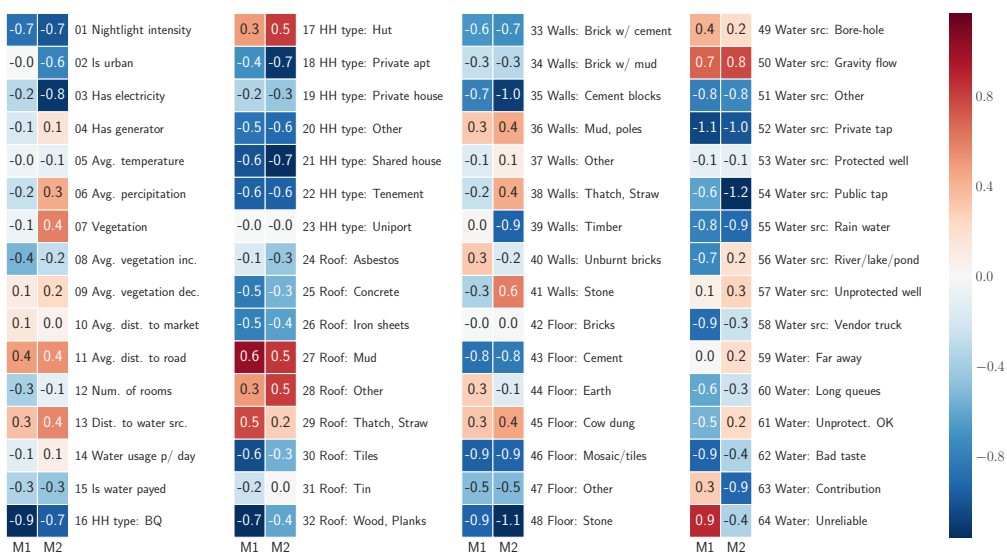

(a) Full visualization of models M1 and M2 learned by CEN on Satellite data.

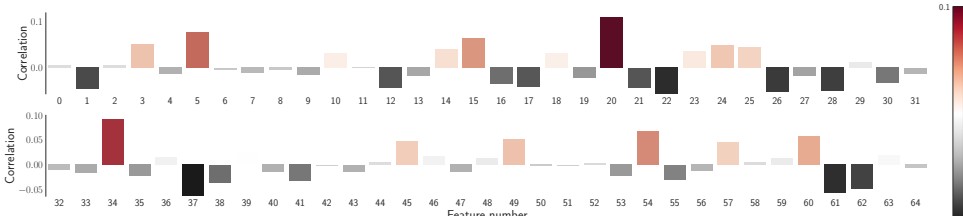

(b) Correlation between the selected explanation and the value of a particular survey variable.

Figure 13: Additional visualizations for CENs trained on the Satellite data.

Table 3: Top-performing architectures used in our experiments on MNIST and IMDB datasets.

(a) MNIST

| Convolutional Encoder | | Contextual Explanations | |
|---|---|---|---|
| layer | Conv2D | model | Logistic regr. |
| # filters | 32 | features | HOG (3, 3) |
| kernel size | $3 \times 3$ | # of features | 729 |
| strides | $1 \times 1$ | standardized | Yes |
| padding | valid | dictionary | 256 |
| activation | ReLU | $l_1$ penalty | $5 \cdot 10^{-5}$ |
| layer | Conv2D | $l_2$ penalty | $1 \cdot 10^{-6}$ |
| # filters | 32 | model | Logistic reg. |
| kernel size | $3 \times 3$ | features | Pixels (20, 20) |
| strides | $1 \times 1$ | # of features | 400 |
| padding | valid | standardized | Yes |
| activation | ReLU | dictionary | 64 |
| layer | MaxPoo2D | $l_1$ penalty | $5 \cdot 10^{-5}$ |
| pooling size | $2 \times 2$ | $l_2$ penalty | $1 \cdot 10^{-6}$ |
| dropout | 0.25 | **Contextual VAE** | |
| layer | Dense | prior | Dir(0.2) |
| units | 128 | sampler | LogisticNormal |
| dropout | 0.50 | | |
| # of blocks | 1 | | |
| # params | 1.2M | | |

(b) IMDB

| Squential Encoder | | Contextual Explanations | |
|---|---|---|---|
| layer | Embedding | model | Logistic reg. |
| vocabulary | 20k | features | BoW |
| dimension | 1024 | # of features | 20k |
| layer | LSTM | Dictionary | 32 |
| bidirectional | Yes | $l_1$ penalty | $5 \cdot 10^{-5}$ |
| units | 256 | $l_2$ penalty | $1 \cdot 10^{-6}$ |
| max length | 200 | model | Logistic reg. |
| dropout | 0.25 | features | Topics |
| rec. dropout | 0.25 | # of features | 50 |
| layer | MaxPool1D | Dictionary | 16 |
| # params | 23.1M | $l_1$ penalty | $1 \cdot 10^{-6}$ |
| | | $l_2$ penalty | $1 \cdot 10^{-8}$ |
| | | **Contextual VAE** | |
| | | Prior | Dir(0.1) |
| | | Sampler | LogisticNormal |

Table 4: Top-performing architectures used in our experiments on CIFAR10 and Satellite datasets. `VGG-16` architecture for CIFAR10 was taken from `https://github.com/szagoruyko/cifar.torch` but implemented in Keras with TensorFlow backend. Weights of the pre-trained `VGG-F` model for the Satellite experiments were taken from `https://github.com/nealjean/predicting-poverty`.

(a) CIFAR10

| Convolutional Encoder | | Contextual Explanations | |
|---|---|---|---|
| model | VGG-16 | model | Logistic reg. |
| pretrained | No | features | HOG (3, 3) |
| fixed weights | No | # of features | 1024 |
| reference | Zagoruyko, 2015 | dictionary | 16 |
| | | $l_1$ penalty | $1 \cdot 10^{-5}$ |
| layer | Dense | $l_2$ penalty | $1 \cdot 10^{-6}$ |
| pretrained | No | **Contextual VAE** | |
| fixed weights | No | prior | Dir(0.2) |
| units | 16 | sampler | LogisticNormal |
| dropout | 0.25 | | |
| activation | ReLU | | |
| # params | 20.0M | | |

(b) Satellite

| Convolutional Encoder | | Contextual Explanations | |
|---|---|---|---|
| model | VGG-F | model | Logistic reg. |
| pretrained | Yes | features | Survey |
| fixed weights | Yes | # of features | 64 |
| reference | Jean et al., 2016 | dictionary | 16 |
| | | $l_1$ penalty | $1 \cdot 10^{-3}$ |
| layer | Dense | $l_2$ penalty | $1 \cdot 10^{-4}$ |
| pretrained | No | # params | |
| fixed weights | No | **Contextual VAE** | |
| units | 128 | prior | Dir(0.2) |
| dropout | 0.25 | sampler | LogisticNormal |
| activation | ReLU | | |
| # trainable params | 0.5M | | |

Table 5: Top-performing architectures used in our experiments on SUPPORT2 and PhysioNet 2012 datasets.

(a) SUPPORT2

| MLP Encoder | | Contextual Explanations | |
|---|---|---|---|
| layer | Dense | model | Linear CRF |
| pretrained | No | features | Measurements |
| fixed weights | No | # of features | 50 |
| units | 64 | dictionary | 16 |
| dropout | 0.50 | $l_1$ penalty | $1 \cdot 10^{-3}$ |
| activation | ReLU | $l_2$ penalty | $1 \cdot 10^{-4}$ |

(b) PhysioNet Challenge 2012

| Sequential Encoder | | Contextual Explanations | |
|---|---|---|---|
| layer | LSTM | model | Linear CRF |
| bidirectional | No | features | Statistics |
| units | 32 | # of features | 111 |
| max length | 150 | dictionary | 16 |
| dropout | 0.25 | $l_1$ penalty | $1 \cdot 10^{-3}$ |
| rec. dropout | 0.25 | $l_2$ penalty | $1 \cdot 10^{-4}$ |

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
