# OpenReview forum: "Contextual Explanation Networks"
_ICLR.cc/2018/Conference — Reject_

### Official Review · AnonReviewer3 · 2017-11-28
**interesting idea, maybe helpful to differentiate more from previous neural net+graphical models idea**

**Rating:** 6
**Confidence:** 5

**Review:**

the paper is clearly written; it works on a popular idea of combining graphical models and neural nets.

this work could benefit from differentiating more from previous literature.

one key component is interpretability, which comes from the use of graphical models.  the authors claim that the previous art directly integrate neural networks into the graphical models as components, which renders the models uninterpretable. however, it is unclear, following the same logic, why the proposed method has interpretability. after all, how to go from the context to the parameters of the graphical models is still uninterpretable. specifically, it is helpful to pinpoint what is special in this model that makes it interpretable, compared to works like Gao, Y., Archer, E. W., Paninski, L., & Cunningham, J. P. (2016). NIPS or Johnson, M., Duvenaud, D. K., Wiltschko, A., Adams, R. P., & Datta, S. R. (2016). NIPS. also, is there any methodological advancement essential to CENs?

the other idea is to go context specific. this idea has been present in language modeling, for example, amortized embedding models like M. Rudolph, F. Ruiz, S. Athey, and D. Blei (2017). NIPS and L. Liu, F. Ruiz, S. Athey, and D. Blei.  (2017). NIPS. application to medical data is interesting. but it could be helpful for the readers to understand if the idea in this work is fundamentally different from these previous ideas from amortized inference.

a final thing. a common challenge with composing graphical models and neural networks (in interpretable or uninterpretable ways) is that the neural networks will usually eat up all the representational power. the variance captured by graphical models becomes negligible. to this end, the power of graphical models for interpretability is limited. interpretability in this case is not much different from fitting only a neural network, taking the penultimate layer to the output as "context specific features" can claim that we are composing a linear model with a neural network, and the linear model is interpretable. it would be interesting to be clear about how the authors get around this issue.

---

> ### Author Response · Authors · 2017-12-20
> **Thank you for the thoughtful comments and suggestions**
>
> First, we appreciate your pointers to the existing relevant literature that we overlooked / weren’t aware of at the time of submission. We have included these and other recent related work and elaborated the differences between CENs and previous literature in Section 2 (please also see our general comment on the major changes above).
>
> Regarding interpretability of CENs
> ------------------------------------------------
> We agree that interpretability of going from the context to parameters of a graphical model in CEN is a valid concern. Although, there is no silver bullet -- if the data is complex, the model that can accurately represent such data will end up being complex in one way or another. Using neural nets as components of a graphical model (e.g., neural potential functions) results in a powerful model. However, to understand patterns of the relationships between variables of interest learned by such model would require “digging” into each neural component separately.
>
> CENs, on the other hand, manage this complexity by explicitly localizing it in the conditional p(\theta | C) -- once we conditioned on the context of interest, we get an explanation which we can understand by simply inspecting its parameters (see footnote 1 on page 2). In this sense, CENs are akin to modular meta-learning approaches (where one constructs models that generate other models) rather than “monolithic” deep graphical models.
>
> Regarding novelty of our approach, we wish to emphasize that, to the best of our knowledge, we are the first to propose using deep networks for generating parameters for simple graphical models which are then used for prediction and inference. We have elaborated on the differences and similarities in Section 2 in the revised version.
>
> Context representation & amortized inference
> ----------------------------------------------------------------
> CENs assume that the representation of the context is given and fixed; learning context embeddings along with predictions is beyond the scope of this work. But thank you for pointing out relevant work -- in the future work, it would interesting to extend CENs to scenarios where context embeddings [1] are learned jointly with the model, or borrow ideas from context selection [2] to improve interpretability of the map from contexts to explanations.
>
> Amortized inference is unnecessary for the types of CENs considered in this work because p(\theta | C) takes a fairly simple form. If one wishes to have a more hierarchical or structured representation of p(\theta | C), ideas from [3] would be very useful.
>
> Representational power
> ---------------------------------
> Neural networks “eating up” the entire representational power is a valid concern. Dictionary and sparsity constraints were chosen to protect us from such scenarios: by constraining the dictionary size and imposing a small sparsity penalty on its atoms, we force CEN to select explanations from a restricted class of models, and hence implicitly control the representational power available the neural part of the model.
>
> We also wish to emphasize a critical point: the context encoder in CEN does not output “context-specific features” as you point out. Instead, it outputs *parameters* for a graphical model which is then used on top of features X (where X is fixed, not learned). The predictive power of CEN necessarily depends on the quality of X and the class of graphical models that are used as explanations (see Appendix A for a detailed discussion).
>
> ---
> [1] M. Rudolph, F. Ruiz, S. Mandt, and D. Blei (2016). NIPS
> [2] L. Liu, F. Ruiz, S. Athey, and D. Blei. (2017). NIPS
> [3] M. Rudolph, F. Ruiz, S. Athey, and D. Blei (2017). NIPS

---

### Official Review · AnonReviewer1 · 2017-11-30
**An approach for end-to-end learning of interpretable models with experimental support**

**Rating:** 6
**Confidence:** 2

**Review:**

The article "Contextual Explanation Networks" introduces the class of models which learn the intermediate explanations in order to make final predictions. The contexts can be learned by, in principle, any model including neural networks, while the final predictions are supposed to be made by some simple models like linear ones. The probabilistic model allows for the simultaneous training of explanation and prediction parts as opposed to some recent post-hoc methods.

The experimental part of the paper considers variety of experiments, including classification on MNIST, CIFAR-10, IMDB and also some experiments on survival analysis. I should note, that the quality of the algorithm is in general similar to other methods considered (as expected). However, while in some cases the CEN algorithm is slightly better, in other cases it appears to sufficiently loose, see for example left part of Figure 3(b) for MNIST data set. It would be interesting to know the explanation. Also, it would be interesting to have more examples of qualitative analysis to see, that the learned explanations are really useful. I am a bit worried, that while we have interpretability with respect to intermediate features, these features theirselves might be very hard to interpret.

To sum up, I think that the general idea looks very natural and the results are quite supportive. However, I don't feel myself confident enough in this area of research to make strong conclusion on the quality of the paper.

---

> ### Author Response · Authors · 2017-12-20
> **Thank you for the thoughtful comments**
>
> We agree that the qualitative results given in the main text may seem limited, so we have included almost 2 pages of discussion of the additional qualitative results in Appendix F.2 (please also see our general comment on the major changes above).
>
> Regarding your comment on Figure 3b, we don’t think we follow what exactly you mean by that. Figure 3b (left) showcases the decrease in the training error at the early stage of training for a baseline CNN (blue) and two CEN models (green and red) that constructed explanations on different features. CEN-hog attains lower training error faster than the other two models. Performance on the held out test set are given in Table 1. We wish to emphasize that, generally, CENs closely match the performance of the vanilla deep nets when the data is abundant and perform better when the data is scarce.

---

### Official Review · AnonReviewer2 · 2017-12-03
**Interesting approach to combine neural nets and graphical models; can improve by elaborating the contrast with related work**

**Rating:** 6
**Confidence:** 3

**Review:**

The paper proposes an interesting combination of neural nets and graphical models by using a deep neural net to predict the parameters of a graphical model. When the nets are trained on contexts "C" (e.g. satellite images associated with a neighborhood) related to an input "X" (e.g. categorical features describing the neighborhood); and a graphical model relates "X" to targets "Y" (e.g. binary variable encoding poverty level of the neighborhood), then the proposed combination can produce interpretable explanations for its predictions.
This approach compares favorably with post-hoc explanation methods like LIME in experiments on conducted on images (CIFAR-10, MNIST), text (IMDB reviews) and time series (Satellite dataset). The paper is clearly written and might inspire follow-up work in other applications. The description of related work is sparse (beyond a derivation of an equivalence with LIME in some settings, explained in the appendix).
The experiments study interesting effects: what happens when the model relating X and Y is degraded (e.g. by introducing noise into X, or sub-selecting X). The paper can be substantially improved by studying the effect of dictionary size and sparsity regularization more thoroughly.

---

> ### Author Response · Authors · 2017-12-20
> **Thank you for the helpful feedback**
>
> We have elaborated on the contrast between our approach and the related work (please see our general comment on the major changes above).
>
> Dictionary size
> --------------------
> Regarding the effect of the dictionary size, Figure 3a answers this question: when the dictionary size is very small, CENs behave almost like the linear models (in the extreme case when the dictionary size is 1, CEN becomes equivalent a single linear model). Larger dictionaries allow for more flexibility so that CENs approach or surpass the performance of the vanilla deep networks.
>
> Sparsity regularization
> ------------------------------
> We found that the results were quite stable and not very sensitive to the sparsity regularization hyperparameter. Adding a small sparsity penalty on the dictionary (between 1e-6 and 1e-3) helped to avoid overfitting for very large dictionary sizes -- the model learned to use only a few explanations from the dictionary for prediction while shrinking the rest of the dictionary to zeros. For instance, on the Satellite data, we set the dictionary size to an arbitrary number (16 or 32) and the model learned to always select between only 2 explanations (M1 and M2) and kept using those for prediction.
>
> We have elaborated on these two points in Section 4.1.1 in the revised version.

---

### Author Response · Authors · 2017-12-20
**Major updates**

We would like to thank all the reviewers for their time and valuable feedback.

We have updated our submission to address reviewer’s concerns and suggestions. Here we detail the major changes that have been made to the manuscript. We answer specific questions raised in the reviews by separately replying to each of them.

Related work
------------------
We have extended the related work (Section 2) and (i) elaborated the key differences between the existing work and contextual explanation networks, (ii) included the related recent work suggested by the reviewers. Additionally, we have extended conclusion (Section 6) with a brief discussion of the limitations of our method and potentially interesting avenues for future work, again connecting CENs to the existing literature.

Qualitative analysis
--------------------------
Qualitative results for MNIST and IMDB datasets were originally included in Figures 9, 10, 11, 12, 13 in the appendix. In the updated version, we further added a detailed discussion of each of these qualitative results in Appendix F.2. To keep the length of the main text under a reasonable page limit, we restricted qualitative analysis in the main text to only one of the applications.

---

### Decision · Program_Chairs · 2018-01-29
**ICLR 2018 Conference Acceptance Decision**

**Decision:**

Reject

**Comment:**

The paper proposes a method to learn and explain simultaneously. The explanations are generated as part of the learning and in some sense come for free. It also goes the other way in that the explanations also help performance in simpler settings. Reviewers found the paper easy to follow and the idea has some value, however, the related work is sparse and consequently comparison to existing state-of-the-art explanation methods is also sparse. These are nontrivial concerns which should have been addressed in the main article not hidden away in the supplement.